ⓐ | **Open Peer Review** | Environmental Microbiology | Methods and Protocols

# Integrating taxonomic and phenotypic information through FISH-enhanced flow cytometry for microbial community dynamics analysis

Valérie Mattelin,[1] Josefien Van Landuyt,[1] Frederiek-Maarten Kerkhof,[2] Yorick Minnebo,[1] Nico Boon[1,2,3]

**ABSTRACT** Flow cytometry is a powerful tool to monitor microbial communities, as it allows tracking both changes in the subpopulations and cell numbers at high throughput and a low sample cost. This information can be combined in a phenotypic fingerprint that can be leveraged for diversity analysis. However, as isogenic individuals can manifest phenotypic diversity, for example, due to differing physiological state and phenotypic plasticity, combining the phenotypic information with taxonomic information adds an extra dimension for describing the dynamics of a microbial community. In this research, taxonomic information was incorporated in the microbial fingerprint through fluorescent *in situ* hybridization (FISH) at a single-cell level. To validate this concept and explore its versatility, two ecosystems with different micro-biodiversity were considered. In the first environment, marine bacteria were monitored for plastic biodegradation in a trickling filter, and in the second, an *in vitro* simulated human gut microbiome was followed over time. Samples were prepared using different (staining) methods, including FISH, and beta diversity analysis was used to evaluate the level of distinction between differently treated groups in both environments. As a reference to correlate increased distinction with the incorporation of taxonomic information, 16S rRNA gene sequencing was used. Finally, a predictive algorithm was trained to correctly classify samples in the differently treated groups. The results showed that the implementation of FISH in flow cytometry provides more information on a single-cell level to answer specific scientific questions, like distinguishing between phenotypically similar communities or following a specific taxonomic group over time.

**IMPORTANCE** Understanding microbial communities is crucial for elucidating their role in maintaining ecosystem health and stability. Researchers are increasingly interested in studying microbial communities by looking at not just their genetic makeup but also their physical traits and functions. In our study, we used common techniques like fluorescence *in situ* hybridization and flow cytometry, along with advanced data analysis, to better understand these communities. This combination allowed us to gather and use data more effectively, demonstrating that these easy-to-use methods, when paired with proper analysis, can enhance our understanding of changing microbial ecosystems.

**KEYWORDS** phenotypic fingerprinting, flow cytometry, fluorescence *in situ* hybridization, microbial community dynamics

Address correspondence to Nico Boon, Nico.Boon@UGent.be.

Valérie Mattelin and Josefien Van Landuyt contributed equally to this article. The author order was determined both alphabetically and in order of increasing seniority.

The authors declare no conflict of interest.

See the funding table on p. 17.

As microbial communities can undergo rapid succession, the composition of microbial communities can change quickly in response to environmental changes. Therefore, rapid methods for accurate analysis of microbial community dynamics are required (1–4). It has been established that a combination of taxonomic and phenotypic information is the most suitable for describing the dynamics of a microbial community,

to compare different microbial communities, or to investigate functional properties, as genetically uniform individuals can manifest phenotypic diversity (3, 5, 6).

Flow cytometry (FCM) can be a fast tool that is extremely suitable for absolute quantification and uncovering the phenotypic heterogeneity of communities. On the other hand, genotypic information has, up to now, been obtained mostly by more time-consuming and expensive (omics) methods, prone to technical biases, such as extraction and PCR bias, and without absolute microbial abundance quantification (3, 4). However, more recently, big strides have been made to overcome PCR bias and time constraints, with, for example, the Oxford Nanopore Technology (ONT), where, in theory, metagenome sequencing of the community can be performed within a day's work without amplification. Linking taxonomic information to phenotypic information to obtain absolute taxon abundances per phenotype on a population level would still be a time-consuming and tedious work (e.g., a combined strategy of cell counting, cell sorting, subsequent 16S rRNA gene amplicon sequencing, and data analysis can take several months of work), although successfully applied in the past (7).

Flow cytometry is a rapid, inexpensive, and robust technique generating real-time quantitative multiparametric data (3, 4, 8). Using general nucleic acid dyes, information on both genome size (cells with high nucleic acid or low nucleic acid contents) (1) and cell viability can be obtained (4). Combined with cell size, morphology, and internal structure (forward scatter and side scatter) features, these large parametric data sets can indicate significant interspecies variation (in cell structure and physiology) (3, 4). By dividing the parameter space into regions and reducing the data to cell abundance per cluster, a so-called phenotypic fingerprint for the bacterial community is generated (2–4, 8–10). This fingerprint is indicative of taxonomic changes in microbial community composition dynamics. For example, it has been used to determine the Crohn's disease state and to monitor drinking water quality (6, 11). In addition, it can be deployed for classical diversity analysis based on diversity indices like the Hill numbers (3, 4, 12).

Despite the successful and versatile applications in various fields (6, 7, 11, 13) and the practical advantages (fast, low-cost, and high-throughput) of flow cytometry, incorporating taxonomic information might increase the resolving power of the microbial fingerprint. Fluorescent *in situ* hybridization (FISH), an rRNA-based method, has been successfully used in combination with flow cytometry before (8, 14–17). The so-called FCM-FISH technique has been established as an effective, versatile, and rapid method to evaluate, for example, the microbial composition of gut microbiota (18–20) or follow specific species over time in simulated gut experiments (21, 22).

An additional FISH labeling (on top of conventional nucleic acid staining) allows a direct link of taxonomic (FISH), translational (rRNA-based FISH), and phenotypic (FCM) information to obtain an integrated fingerprint of the community, all in one single measurement. The translational information obtained by FISH is linked to the potential translational activity represented by the number of ribosomes in the cell (16). We hypothesize that the addition of fluorescent rRNA probes would increase the resolving power of the phenotypic fingerprint, enabling the discovery of community shifts in global community structure that would have gone unnoticed with nucleic acid (NA) staining alone. Fluorescently labeling specific taxonomic groups not only add an extra dimension to the fingerprint but they could also help link the phenotypic behavior of the microbial community to particular taxa. Consequently, phenotypic fingerprinting could be performed per taxonomic group, gathering single-cell information of this specific group within a population.

The combined technique could enable us to monitor bacterial responses to specific process variations, focusing on taxonomic groups with significant functional roles. This approach allows for a systematic comparison with 16S rRNA gene sequencing to validate its effectiveness. To validate this hypothesis, two different systems were used, one in which enriched marine bacteria were monitored for plastic biodegradation in a trickling filter bioreactor design (23) and a second in which a gut microbial community was simulated *in vitro* in the Simulator of the Human Intestinal Microbial Ecosystem

(SHIME) (24–26). FISH probe efficiency was confirmed; the microbial beta-diversity and its dynamics in the different ecosystems were calculated; and accuracy for sample-level classification was assessed compared to 16S rRNA amplicon sequencing. The results show that the described technique is versatile and offers great promise for microbial ecological monitoring.

## MATERIALS AND METHODS

### Sampling

Sample collection and experimental setups of both ecosystems are described in the supporting information (SI). In short, marine ecosystem samples were collected from the effluent of four trickling filter bioreactors run with artificial seawater medium (in S1) and inoculated with fresh coastal seawater: two with PHBH plastic (referred to as P) and two with a novel plastic, B4PF01 (referred to as F), to investigate plastic biodegradation. Effluent samples for fixation and flow cytometry were collected thrice weekly for 56 days. Samples for 16S rRNA gene amplicon sequencing were collected on days 0, 14, 28, 35, 42, and 56 (23). *In vitro* simulated gut microbiome samples were derived from the Simulator of the Human Intestinal Microbial Ecosystem (SHIME, Prodigest, Zwijnaarde, Belgium). The SHIME model simulated the proximal and distal colon in different, but consecutive, vessels, with controlled pH, residence time, temperature (37°C), mixing (200 rpm), and diet (25–27). Two setups were used, each in duplicate. The first was a standardized SHIME with fixed eating patterns, transit time, and nutritional media. The second was an individualized SHIME with adjusted parameters, according to fecal donor characteristics. Fecal donor samples were inoculated in the colon vessels, and the system stabilized over 11 days. Samples were collected on days 1, 2, 4, 7, 9, and 11 for analysis (FCM and 16S rRNA gene amplicon sequencing) (24).

### Fixation

Fixation was performed in two ways: as both gram-positive, preferably fixed in ethanol, and gram-negative, preferably fixed in paraformaldehyde (PFA), microorganisms were the target in this study. For gram-positive bacteria, the sample was added to a 1:1 sample:ethanol mixture and stored at −20°C. For gram-negative bacteria, the sample was added in 1:3 sample:paraformaldehyde buffer solution (4% diluted PFA (Merck, Darmstadt, Germany) in phosphate-buffered saline (PBS) (130 mM NaCl; 5 mM $Na_2HPO_4.2H_2O$; 5 mM $NaH_2PO_4.2H_2O$; adjusted pH 7.2)) overnight, washed in PBS solution, and stored at −20°C in 1:1 PBS:ethanol mixture.

### Fluorescent *in situ* hybridization

The protocol for FISH was based on Huang et al. (28). Between 50 and 200 µL of each fixed sample was spotted into a MultiScreen$_{HTS}$-GV (0.22 µm filter) 96-well plate (Merck, Darmstadt, Germany) and pulled through a MultiScreen Vacuum Manifold 96-well (Merck). Subsequently, all samples were resuspended in 100 µL 100% grade ethanol and incubated for 5 min at room temperature for dehydration. Ethanol was removed by vacuum filtration, and the cells were air-dried. Cells were hybridized in hybridization buffer (100 µL) for 3 h at 46°C. The hybridization buffer consisted of 900 mM NaCl, 20 mM TRIS HCl, 0.5 M EDTA, 0.01% SDS, and contained 100 ng/µL of the respective fluorescently labeled oligonucleotide probes as well as the required formamide concentration (20% for the marine samples and 30% for the SHIME samples) to obtain stringent conditions. After hybridization, samples were vacuum filtered to remove buffer and oligonucleotide probe and resuspended in pre-heated wash buffer (48°C) of appropriate stringency and incubated for 15 min at 48°C. The wash buffer contained 0.5 mM TRIS HCl, 5M NaCl, 0.5 M EDTA, 10% SDS, and MilliQ (Metrohm, Kontich, Belgium). Finally, the wash buffer was filtered off, and the cells were resuspended in the original volume used (50–200 µL PBS). Negative control samples were included (sterile PBS).

Established FISH probes were chosen based on their ability to bind with taxonomic groups present in the samples, binding effectiveness in literature and/or presence in Probebase (29) (Table S1). As a control, the probes for all bacteria, a mixture of EUBI, EUBII, and EUBIII probes in a ratio of 1:1:1 (EUBmix), were used. The complete proposed workflow is shown in Fig. 1.

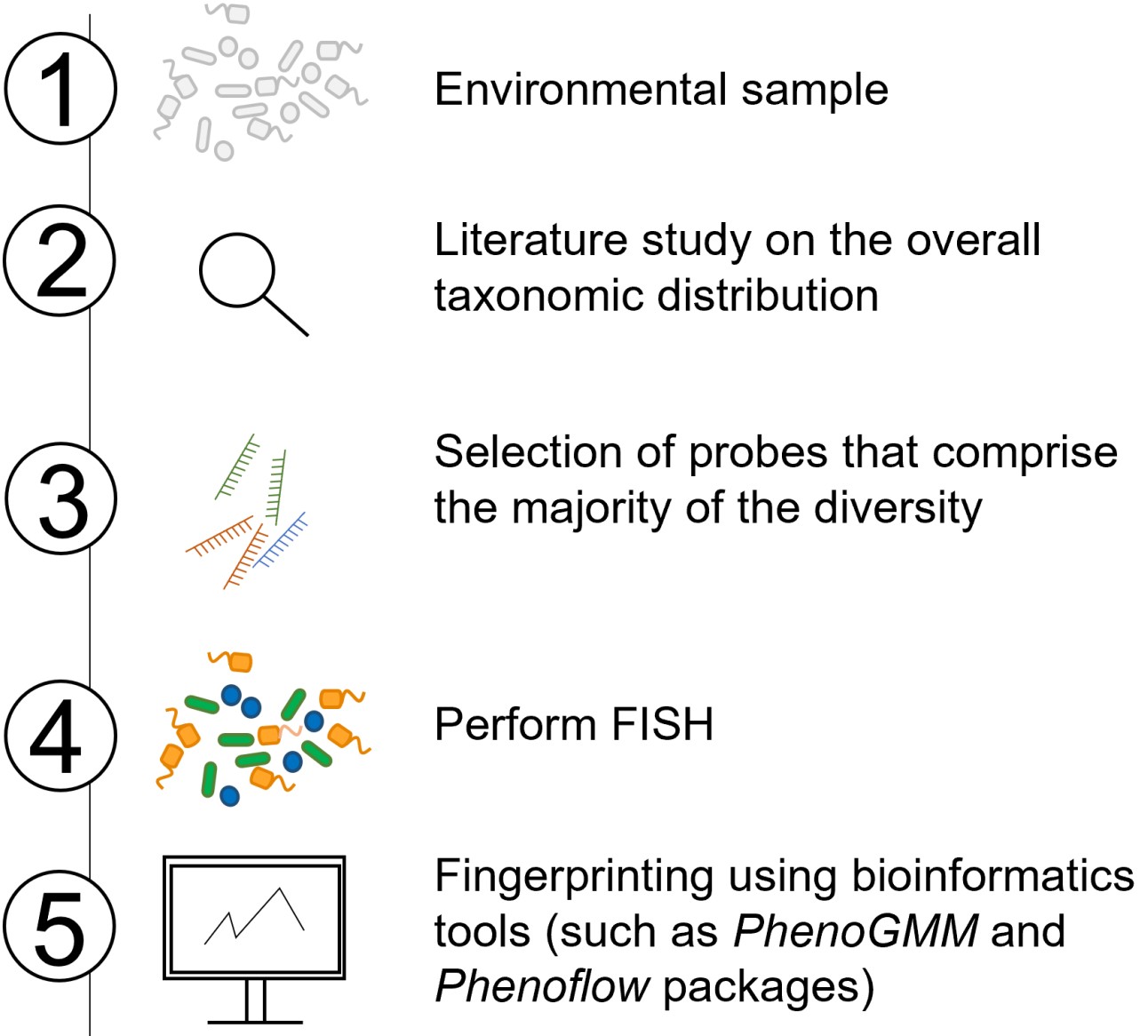

# FISH-Flow cytometry fingerprinting (FFCP)
## A new way of taxonomy based fingerprinting

1. Environmental sample

2. Literature study on the overall taxonomic distribution

3. Selection of probes that comprise the majority of the diversity

4. Perform FISH

5. Fingerprinting using bioinformatics tools (such as *PhenoGMM* and *Phenoflow* packages)

**FIG 1** Proposed workflow when applying FISH-enhanced flow cytometry starting from (1) (environmental) mixed-culture samples (2). Screening the overall taxonomic structure of the samples based on literature (3), followed by a selection of probes covering the most important taxa (4), performing FISH and flow cytometry measurements, after which (5) packages such as *Phenoflow* (4) and *PhenoGMM* (30) can be applied. Modified from Mermans et al. (31).

## Flow cytometry analysis

Fresh samples were diluted to average concentrations of $10^6$–$10^7$ cells/mL in 0.22 µm-filtered aerobic or anaerobic PBS buffer (Table S2) for the marine and SHIME ecosystem, respectively. SYBR Green I (SG) (10,000× concentrate in DMSO) (Invitrogen, Eugene, USA) was diluted 100 times in 0.2 µm filtered DMSO (Merck), propidium iodide (PI) (Invitrogen, Eugene, USA) and 4′,6-diamidino-2-phenylindole (DAPI) (Merck) were diluted in 0.2 µm filtered DMSO down to 0.4 mM and 3 mM, respectively. Samples were subsequently stained with 1% vol/vol SG or SGPI. Fixed-FISH samples were diluted to average concentrations of $10^6$–$10^7$ cells/mL in 0.22 µm-filtered PBS solution and stained either with 1% vol/vol SG or with 1% vol/vol DAPI. After staining, all samples were incubated for 20 min at 37°C in the dark prior to analysis (32). Samples were analyzed immediately after incubation on a BD FACSVerse Cell Analyzer (BD Biosciences) equipped with a violet (405 nm, detectors 448/15 and 528/45 nm), a blue (488 nm, detectors 527/32 and 700/54 nm), and red (637 nm, detectors 660/10 and 783/56 nm) laser with BD FACSFlow Sheath Fluid (BD Biosciences) as sheath fluid. The instrument performance was verified daily using BD FACSuite CS&T research beads (BD Biosciences). Flow rate was set at "medium" (±50 µL/min), and photomultiplier tube voltages were adjusted to have optimal detection of the selected fluorophores (more details in SI). Negative (0.2 µm filtered samples) and positive (heat-killed samples for membrane-compromised control) controls were run along with the samples to define gates for filtering cells from debris and separating membrane-intact and membrane-damaged cells (Fig. S1).

## 16S rRNA gene amplicon sequencing

DNA from the marine ecosystem samples was extracted by means of bead beating with a PowerLyzer instrument (MoBio) and phenol/chloroform extraction as described by Van Landuyt et al. (33). The extraction was followed by PCR with universal bacterial primers (341F (5′-CCT ACG GGN GGC WGC AG-3′) and 785Rmod (5′-GAC TAC HVG GGT ATC TAA KCC-3′) targeting V3-V4 region of the 16S rRNA gene and sent to LGC Genomics for Illumina. DNA from the SHIME ecosystem samples was extracted and sequenced according to Vandeputte et al. (34).

## Data analysis

Data analysis was performed using the R language for statistical programming with the add-on packages *phyloseq* (v1.22.3) (35), *vegan* (v2.5.6) (36), *Phenoflow* (v1.1.2) (4), and *caret* (v 6.0-92) (37). All data analysis was performed in R (version 4.0.3) (38).

### 16S rRNA gene amplicon sequencing data analysis

The DADA2 R package was used to process the amplicon sequence data according to the pipeline tutorial (39), as described by Van Landuyt et al. (40). Copy number correction was performed using average copy number per assigned family retrieved from the rrnDB database (41).

### Fingerprinting of flow cytometry data and diversity analysis

Fingerprinting was performed with the PhenoGMM function integrated in *Phenoflow* (v1.1.2) (4) further developed in Rubbens et al. (10). Samples were downsampled to 6,000 cells for model building. The number of mixtures was set to 30, based on the optimal number of mixtures determined by the Bayesian Information Criterion (BIC, VVV: varying volume, varying shape, varying orientation [ellipsoidal covariance]). The models were generated based on the SSC-H, FSC-H, B527-H, B700-H, R660-H, R783-H, V448-H, and V528-H detector channels and were applied to the flow cytometry data by the PhenoMaskGMM() function to determine the events per mixture for each sample.

Distance matrices were computed by the Bray-Curtis dissimilarity, with the vegdist() function from *vegan* (v2.5.6) (36). Ordination by PCoA, NMDS, and tSNE was performed

on the dissimilarity matrices. The multivariate data were analyzed for differences in group discrimination within both the marine and the human ecosystem by calculating the variability within each group using a multivariate analog of Levene's test (*betadisper*() from *vegan*), testing the statistical significance of this variability using a permutation-based test of multivariate homogeneity, and by pairwise comparison of the distance between the group centroids using Permutational Multivariate Analysis of Variance (PERMANOVA) (*adonis*() and *adonis2*() functions from *vegan*), using a cut-off of (adjusted) *P*-value > 0.05. The Bray-Curtis distance matrices of the flow cytometry data were correlated to the Bray-Curtis distance matrix of the 16S rRNA gene amplicon sequencing data by calculating the Mantel statistic (*mantel*() from *vegan*) and using the Spearman non-parametric correlation.

### Training a predictive algorithm

A predictive model was trained using the *caret* (v 6.0-92) (37) package. Each data set (i.e., the different staining methods in both ecosystems) was partitioned into a training and a testing data set (ratio 80:20) by fivefold nested cross-validation. A random forest model was trained to classify the samples in the correct group, using 99 bootstraps, with the *train*() and *trainControl*() functions. The random forest algorithm was chosen as it was shown to be a reliable method for community composition in synthetic communities (42, 43). The models were evaluated using the area under the ROC curve (AUC), *pROC* (v 1.18-4) package (44).

## RESULTS

### Fingerprinting and diversity analysis

### Distinction between marine microbial communities, grown on different plastic types, is superior using FISH

Marine, plastic-enriched communities were grown in two separate trickling filter bioreactors, with two different plastic types (P and F as explained above), in duplicate, to identify whether different plastics select for other microbial communities. Using four different staining methods, we investigated whether the flow cytometric fingerprinting could be improved: (i) non-fixed samples stained with Sybr Green (SG), (ii) paraformaldehyde (PFA) fixed samples stained with SG, (iii) fixed samples stained with 4′,6-dia-midino-2-phenylindole (DAPI), and (iv) fixed FISH samples counterstained with DAPI. Diversity analyses of these were compared with the diversity analysis of the 16S rRNA gene amplicon sequencing samples.

Principal coordinates analysis (PCoA) was performed to distinguish between microbial community fingerprints (based on Gaussian Mixture Models (GMM)) growing on the two types of plastics (F vs. P). The microbial communities in the reactors with the two different plastics as a carbon source were segregated for the fixed FISH stained samples (on the first two ordination axes), indicated by the ellipse drawn on a 95% confidence level (Fig. 2E), which is in line with the results for the 16S rRNA gene amplicon sequencing data (Fig. 2A). Within the non-fixed, SG stained flow cytometry data, there was a clear visual difference between the inoculum samples and the later timepoints (Fig. 2B), which was not visible when plotting the 16S rRNA gene amplicon sequencing data, nor in the other staining methods. This is also observed in the (fluorescence-based) scatterplots of the data (Fig. S2). The fixed samples, stained with either SG or DAPI, could not discriminate between both the plastic-type groups when ordinated based on PCoA. Ordination based on NMDS and dimension reduction based on tSNE was performed, with similar results (Fig. S3 and S4).

The different staining methods and the discriminatory potential were compared by the size of the variation of samples of the same group (i.e., size of the sample groups, here separated based on plastic type) and the distance between the centers of each group (i.e., centroids of the sample group) by statistical tests for multivariate data. The variance in each group was calculated to ensure homoscedasticity prior

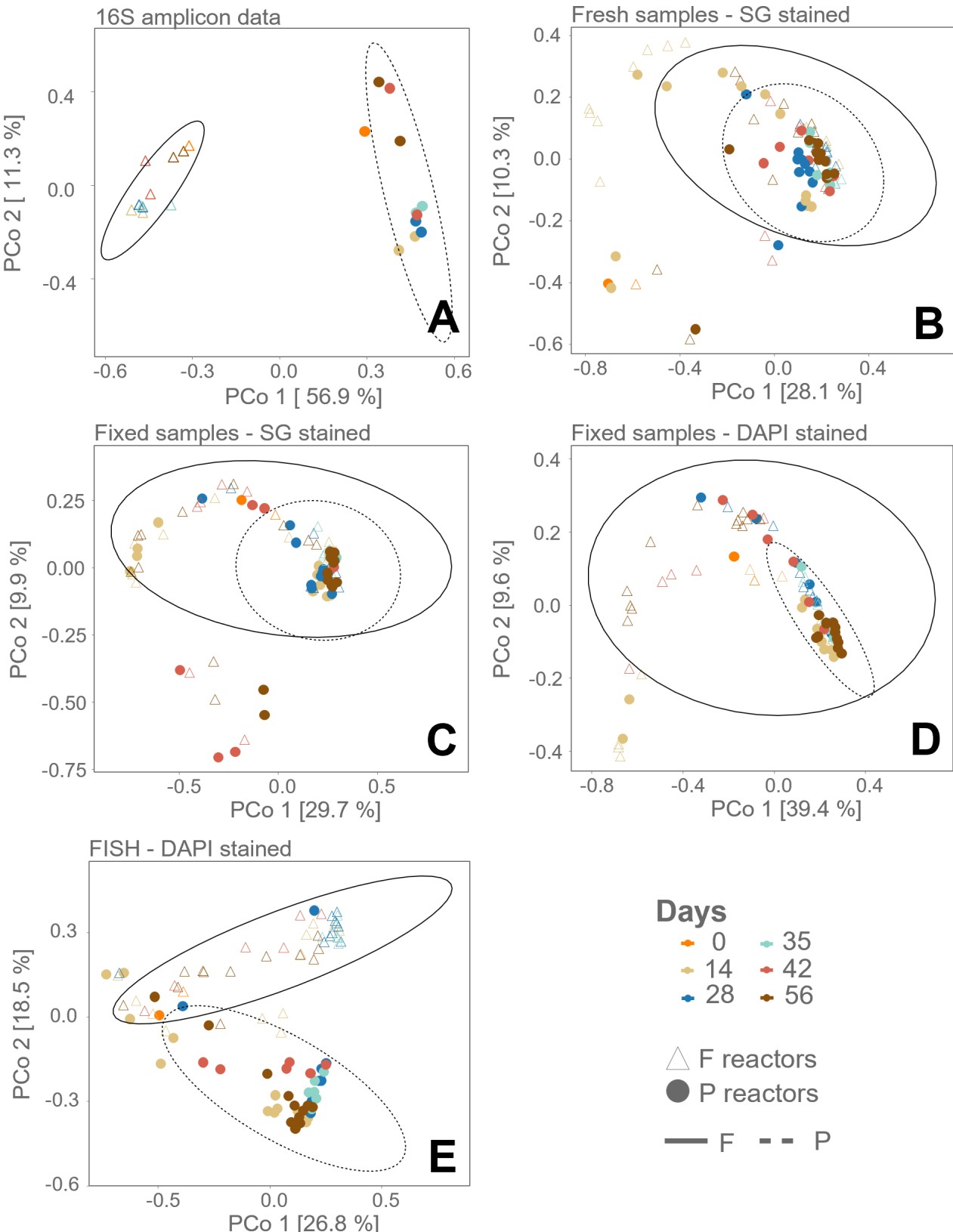

**FIG 2** PCoA diversity analysis of the marine trickling filter samples, based on Bray-Curtis dissimilarity matrix calculated from (A) 16S rRNA gene amplicon sequencing, (B) SG staining of live cells, (C) SG staining of PFA-fixed cells, (D) DAPI staining of PFA-fixed cells, and (E) FISH and DAPI staining of PFA-fixed cells. The different reactors, based on plastic material (P: PHBH-fed, F: B4PF01-fed), are displayed by empty (○) and full (●) circles. The ellipse is drawn on the 95% confidence level.

to PERMANOVA testing. The null hypothesis (i.e., no difference between groups) was accepted for every staining method, except the DAPI-stained fixed samples and the 16S rRNA gene amplicon sequencing samples. The distances between the means of the groups (i.e., centroids) were calculated and tested by PERMANOVA and found to be significantly different for each staining method (Table S2). However, the distance between the centroids (distance = 0.32, $P$ = 0.001), as well as the pseudo $F$-ratio (F = 17.56, $P$ = 0.001), the so-called effect size of the centroid distance comparison through PERMANOVA analysis, was higher for the FISH samples compared to all three other staining methods, namely the non-fixed samples (distance = 0.15, $P$ = 0.006; F = 3.98, $P$ = 0.005), the fixed samples stained with SG (distance = 0.19, $P$ = 0.004; F = 4.72, $P$ = 0.001), and the fixed samples with DAPI (distance = 0.20, $P$ = 0.002; F = 9.70, $P$ = 0.001; ANOSIM statistic 0.108, $P$ = 0.001), confirming statistically what was visualized in Fig. 2, that is, that the FISH stained samples gave a better discrimination (Table S2).

The Bray-Curtis distance matrices of the flow cytometry data were correlated to the Bray-Curtis distance matrix of the 16S rRNA gene amplicon sequencing data with the Mantel test, using the Spearman non-parametric correlation test. Although all staining methods are significantly correlated, the highest significant correlation was found for the fixed FISH samples (R = 0.4559, $P$ = 0.0001). The non-fixed, SG-stained samples had the second highest significant correlation (R = 0.2242, $P$ = 0.0001) (Table 1). Note that the sample sizes of the flow cytometry data (94 samples/staining method) were reduced to the sample size of the sequencing data set (22 samples; the corresponding samples were taken), which influences the diversity estimations (both statistical comparisons and Mantel test).

## Microbial ecology in a human microbiome simulation

The *in vitro* simulated gut microbiomes in SHIME reactors were set up either individualized (with individualized feeding frequencies, media, and transit time based on each fecal donor [SI]) or using the standardized protocol (with fixed feeding frequencies, media, and transit time [SI]), to determine the importance of individualization for future reference (24). Cells were fixed in either EtOH or paraformaldehyde (PFA) and subsequently stained with DAPI and FISH. The PCoA analysis for these different staining methods and the 16S rRNA gene amplicon sequencing data shows that the samples diverge from each other over time, for the PFA, EtOH fixed-FISH labeling method, and the 16S rRNA gene amplicon sequencing data (Fig. 3A, C and D), while there is no clear segregation in the fresh sample staining (both intact-membrane, SG stained, cells and membrane-damaged, PI stained, cells community were used in the PCoA analysis) (Fig. 3B). It is seen that the proximal/individualized diverged further from proximal/standardized vessel by timepoint 11, compared to the distal vessel type (Fig. 3).

The SHIME samples were evaluated using the same statistical approach as the marine bacteria samples. In this case, the 16S rRNA gene amplicon sequencing data, the non-fixed SGPI stained samples, and the ethanol-fixed FISH stained samples showed a significant homoscedasticity for the two groups (based on *betadisper*() and *permutest*(); $P < 0.05$), while the PFA-fixed FISH stained samples did not ($P$ = 0.011) (Table S4). The distance between the centroids, calculated by PERMANOVA, was greater in the non-fixed SGPI stained samples (0.26 vs. 0.22 for FISH PFA; 0.19 for FISH EtOH); however, all were significant (Table S4). The pseudo $F$-ratio, on the other hand, was greater for the PFA

**TABLE 1** Results of the Mantel test, based on the Bray-Curtis dissimilarity matrix, shown by the Pearson correlation coefficient (999 permutations)[a]

| Staining method | Mantel statistic R | Significance |
|---|---|---|
| SG—non fixed | 0.2242 | 0.0001*** |
| SG—fixed | 0.1082 | 0.0404* |
| DAPI—fixed | 0.1214 | 0.0285* |
| FISH—fixed (DAPI) | 0.4559 | 0.0001*** |

[a]The sample size was adapted to be equal to the sample size of the 16S rRNA gene amplicon sequencing ($\alpha$ = 0.05; ***$P$ ≤ 0.001; **$P$ ≤ 0.005; *$P$ ≤ 0.05). Gray highlighting indicates significant $P$ values.

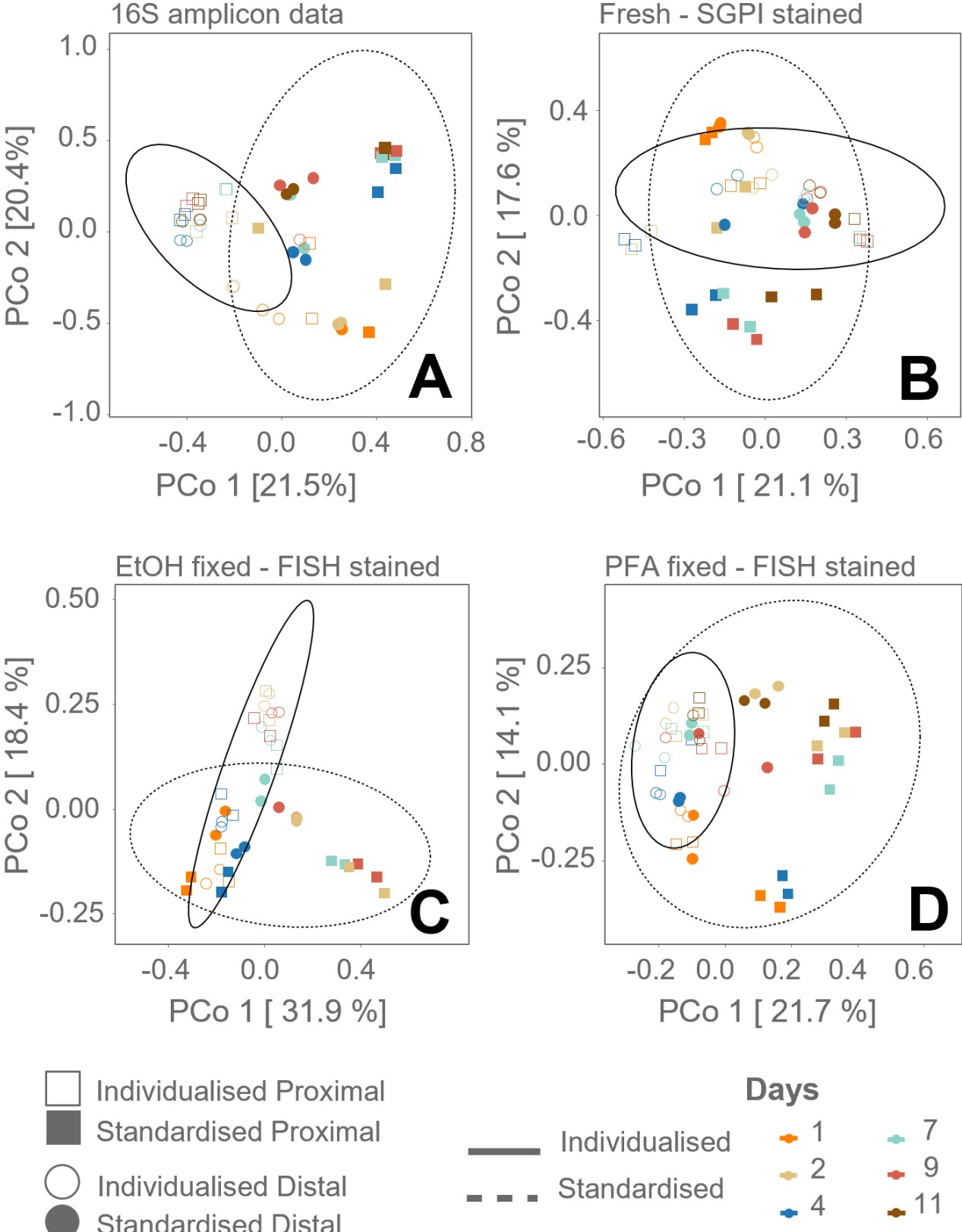

**FIG 3** PCoA diversity analysis of the simulated gut samples, based on Bray-Curtis dissimilarity matrix calculated based on (A) 16S rRNA gene amplicon sequencing, (B) SGPI staining, (C) DAPI and FISH labeling of EtOH-fixed cells, and (D) DAPI and FISH labeling of PFA-fixed cells. The different reactors, based on individualized/standardized SHIME operation and distal/proximal colon vessel, are displayed by empty (○,□) and full (●, ■) circles and squares. Biological duplicates are shown. The ellipse is drawn on the 95% confidence level (based only on SHIME operation settings: individualized and standardized).

fixed-FISH labeling method, containing two probes (F = 11.223, $P$ = 0.001), and the 16S rRNA gene amplicon sequencing data (F = 10.347; $P$ = 0.001), although significant for all staining methods. The lowest effect sizes were obtained using the EtOH fixed-FISH labeling method, containing only one probe ($R^2$=7.15; $P$ = 0.001).

Similar to the marine ecosystem, the 16S rRNA gene amplicon sequencing data of the SHIME experiment were correlated to the flow cytometry staining methods by performing the Mantel test (Spearman correlation, non-parametric). A significant correlation of around 50% between flow cytometry and 16S rRNA gene amplicon sequencing samples for all staining methods was obtained (Table 2). Note that the flow cytometry sample numbers were reduced (day 0 not included due to sequencing failure). Samples stained with FISH (regardless of fixation) showed a better correlation (R = 0.55 and R = 0.59) than non-fixed SGPI stained samples (R = 0.47) (Table 2).

## Predicting the classification into groups by modeling

Since it was hypothesized that more information would be included in an integrated fingerprint, the predictive power of classifying samples was investigated. A training and test data set was created with 80% resp. 20% of the data, five consecutive times. For each training and test data set, a random forest model (43) was trained and built with 99 bootstraps. For the marine ecosystem, the average area under the ROC curve (AUC) was >0.80 for all staining methods, with the highest AUC for the non-fixed SG staining method (0.9854) (Table S4). For the SHIME staining methods, the accuracy was again highest for the FISH-stained samples (1), and both other staining methods have very similar prediction accuracy >0.90 (Table S4).

## FISH labeling of higher-level taxonomic groups

The relative number of FISH-positive cells was compared with the (copy number corrected) relative abundance percentages obtained from 16S rRNA gene amplicon sequencing to get insight into the number of potentially active cells (FISH) within the whole community (16S rRNA gene amplicon sequencing).

### *Marine microbial communities*

FISH labeling was performed on two different taxonomic levels for the plastic-degrading marine microbial communities. First, all bacteria were targeted with a mixture of the three probes: EUBI, EUBII, and EUBIII. When comparing the amount of EUBmix-stained cells (16S rRNA hybridization) to the total amount of DAPI-stained cells (DNA staining), we found that for the reactors, on average, over time, 41.3% ± 14.3% and 46.3% ± 19.2% of all cells were FISH positive for, respectively, both P and both F reactors (Fig. 4).

On a deeper taxonomic level, *Alpha-* and *Gammaproteobacteria* classes were targeted by FISH, as these are two major groups of bacteria present in the marine environment and thus often found in plastic degradation studies (45–48). In total, more than 60% of the community was classified as one of these two classes by 16S rRNA gene amplicon sequencing. Comparing this with the FISH-stained samples, it is shown that the abundances by FISH are consistently lower and comprise a different percentage of the classes over time (Fig. 5). The ratio of *Alpha-* over *Gammaproteobacteria* shows that the sequencing data has a consistent (except for t = 42 days; P) higher ratio. The coverage of the probes was tested *in silico*, allowing one mismatch. The *Alphaproteobacteria*

**TABLE 2** Results of the Mantel test, based on the Bray-Curtis dissimilarity matrix, shown by the Pearson correlation coefficient (999 permutations)[a]

| Sample set | Mantel statistic R | Significance |
|---|---|---|
| SGPI—non fixed | 0.47 | 0.0001*** |
| DAPI-FISH—EtOH fixed | 0.5851 | 0.0001*** |
| DAPI-FISH—PFA fixed | 0.5506 | 0.0001*** |

[a]The sample size of the 16S rRNA gene amplicon sequencing was adapted to be equal to the sample size of the EtOH fixed-FISH samples (α = 0.05; ***$P$ ≤ 0.001; **$P$ ≤ 0.005; *$P$ ≤ 0.05). Gray highlighting indicates significant $P$ values.

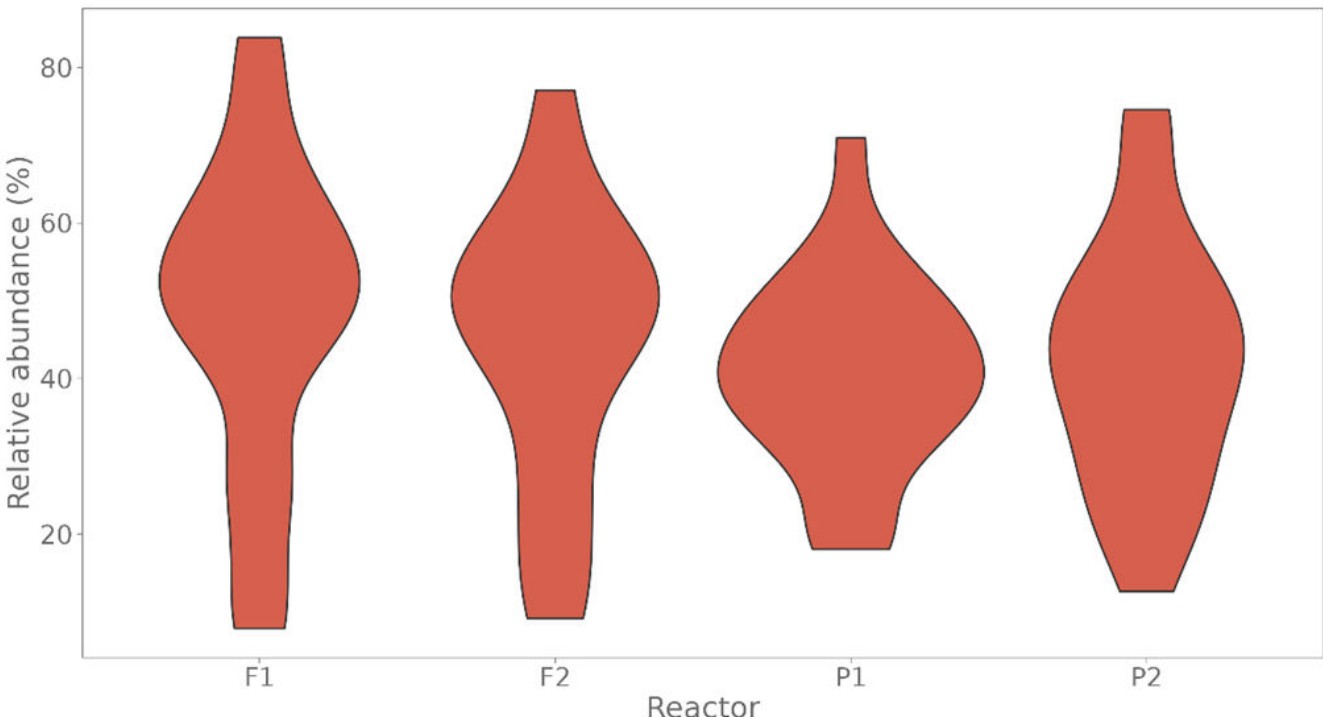

**FIG 4** The relative amount of FISH-positive cells of the marine trickling filter samples (FISH stained by EUBmix, which consists of a mixture of EUBI, EUBII, and EUB III in a ratio of 1:1:1, targeting all cells), compared to the DAPI-stained cells. The different reactors are displayed on the x-axis: communities were grown in two separate trickling filter bioreactors, with two different plastic types, in duplicate (PHBH, referred to as P1 and P2, and a bioplastic, referred to as F1 and F2) (*n* = 24).

probe has 92.77% coverage of *Alphaproteobacteria* sequences (SILVA Small Subunit rRNA database r138.1), the *Gammaproteobacteria* probe 76% of *Gammaproteobacteria* sequences (Long Subunit rRNA database r138.1) and its competitor probe 19.1% (0 mismatches). Within the *Gammaproteobacteria*, 90.1% coverage of the *Pseudomonadales* was found via the SILVA test probe application, an order to which most *Gammaproteobacteria* were classified here.

### Microbial ecology in a human microbiome simulation

The FISH probes were chosen based on the most prevalent phyla in *in vitro* simulated gut microbial ecosystems, namely *Bacillota*, *Bacteroidetes*, and *Gammaproteobacteria* (24). To avoid fixation problems and biases and to identify the best strategy in case of a community with mostly Gram-positive bacteria, the SHIME samples were fixed in two different ways, with both ethanol and PFA. Most *Bacillota* are Gram positive, except the *Veillonella* genus (50 Amplicon Sequence Variants [ASVs] were classified as *Veillonella*), and require EtOH fixation, while *Bacteroidetes* and *Proteobacteria* are Gram-negative bacteria and are preferably fixed in PFA (49, 50).

The FISH labeling of the *Bacteroidota*, *Bacillota,* and the *Gammaproteobacteria* seems to correspond to the translationally active bacteria (Fig. 6A). Remarkably, the abundance of the *Gammaproteobacteria* was very low for the standardized SHIME, compared to the individualized SHIME, visual both in the FISH cell abundances and the taxonomic abundances. Next, it is possible to calculate the ratio between the FISH-positive cells and the 16S rRNA gene amplicon sequencing data to represent the active fraction present in the sample (Fig. 6B). The coverage of the probes was tested *in silico*, allowing one mismatch. The *Bacteroidota* probe has 80.85% coverage of *Bacteroidota* sequences, and the *Bacillota* probes have 57.18% coverage (SILVA Small Subunit rRNA database r138.1).

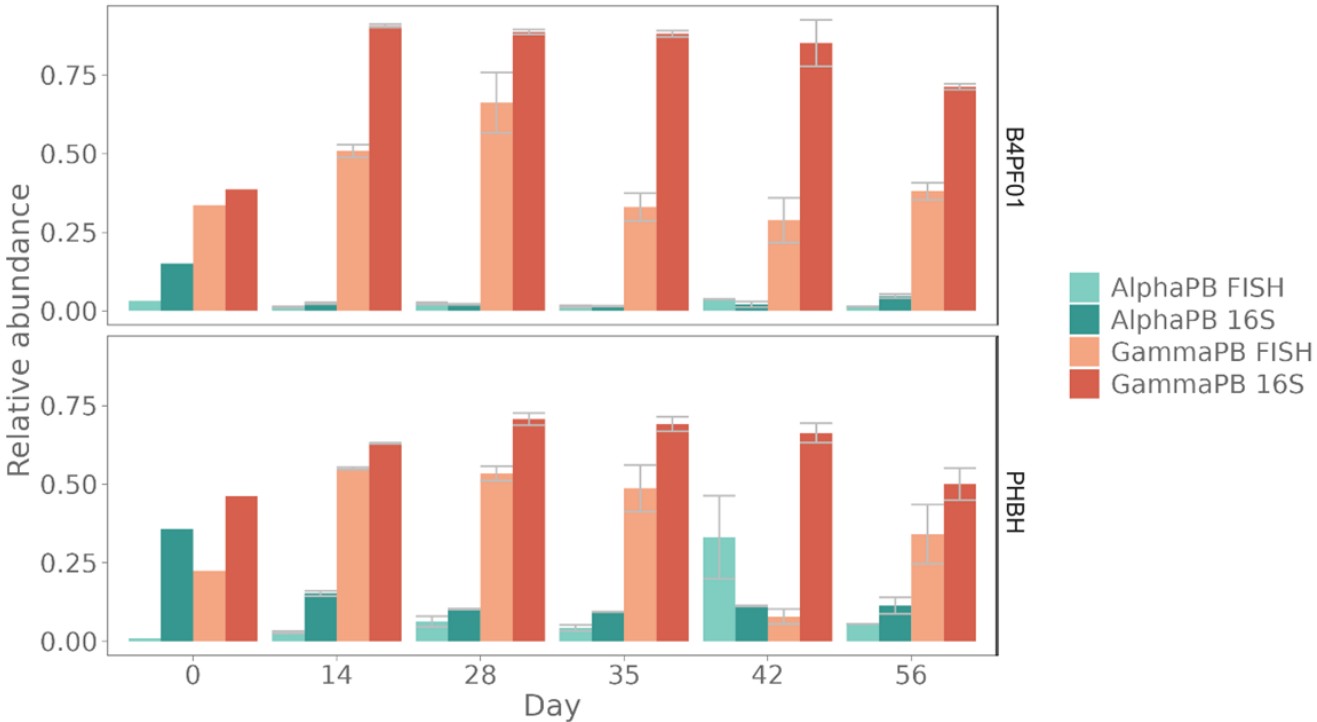

**FIG 5** Comparison of cells of the marine trickling filter samples classified as *Alpha-* (AlphaPB) and *Gammaproteobacteria* (GammaPB) by copy number corrected, identified reads from 16S rRNA gene amplicon sequencing and FISH. Biological duplicate reactors were averaged, and the standard error is shown.

The beta diversity based on the Bray-Curtis dissimilarity matrix of solely the *Bacillota* (FISH) labeled cells was plotted in a PCoA ordination (Fig. 7A) and compared with the beta diversity of all cells from this staining method (Fig. 7B) and the *Bacillota* classified ASVs (Fig. 7C). In our SHIME system, different phases were distinguished, with day 1 being the starting phase, days 2–9 as the stabilization phase, and day 11 as the stabilized phase, in which no fluctuations were noted in the for two consecutive timepoints (as shown in Fig. S7). In the start phase, all different applied conditions (i.e., the diversity of the *Bacillota* across the conditions) plot closely together for all three cases, but the *Bacillota* phenotypic diversity diverges in the stabilization phase according to the applied SHIME condition (individualized/standardized). By day 11, in the stabilized system, the standardized-proximal colon condition can be clearly separated from the other conditions for the 16S rRNA gene amplicon sequencing data. For the *Bacillota* (FISH) labeled cells, a clear separation between individualized and standardized SHIME conditions is seen (Fig. 7B).

## DISCUSSION

In our study, we integrated phenotypic fingerprinting, a technique well-established across various ecosystems (4, 9), with FISH. This combination allowed us to merge physiological, phenotypic, and taxonomic information, providing a more detailed description of the microbial community. Moreover, the implementation of FISH provides information on the ribosomal content (which is a proxy for activity) within the microbial community. This study demonstrates that the implementation of FISH in flow cytometry for microbial diversity analysis provides more information on a single-cell level to answer specific scientific questions.

FISH probes target the rRNA, complementary to the fluorescent DNA probe. As the target is rRNA, some bottlenecks should be considered. The fluorescent signal is dependent on the number of target cells and ribosomes present (i.e., the activity of the

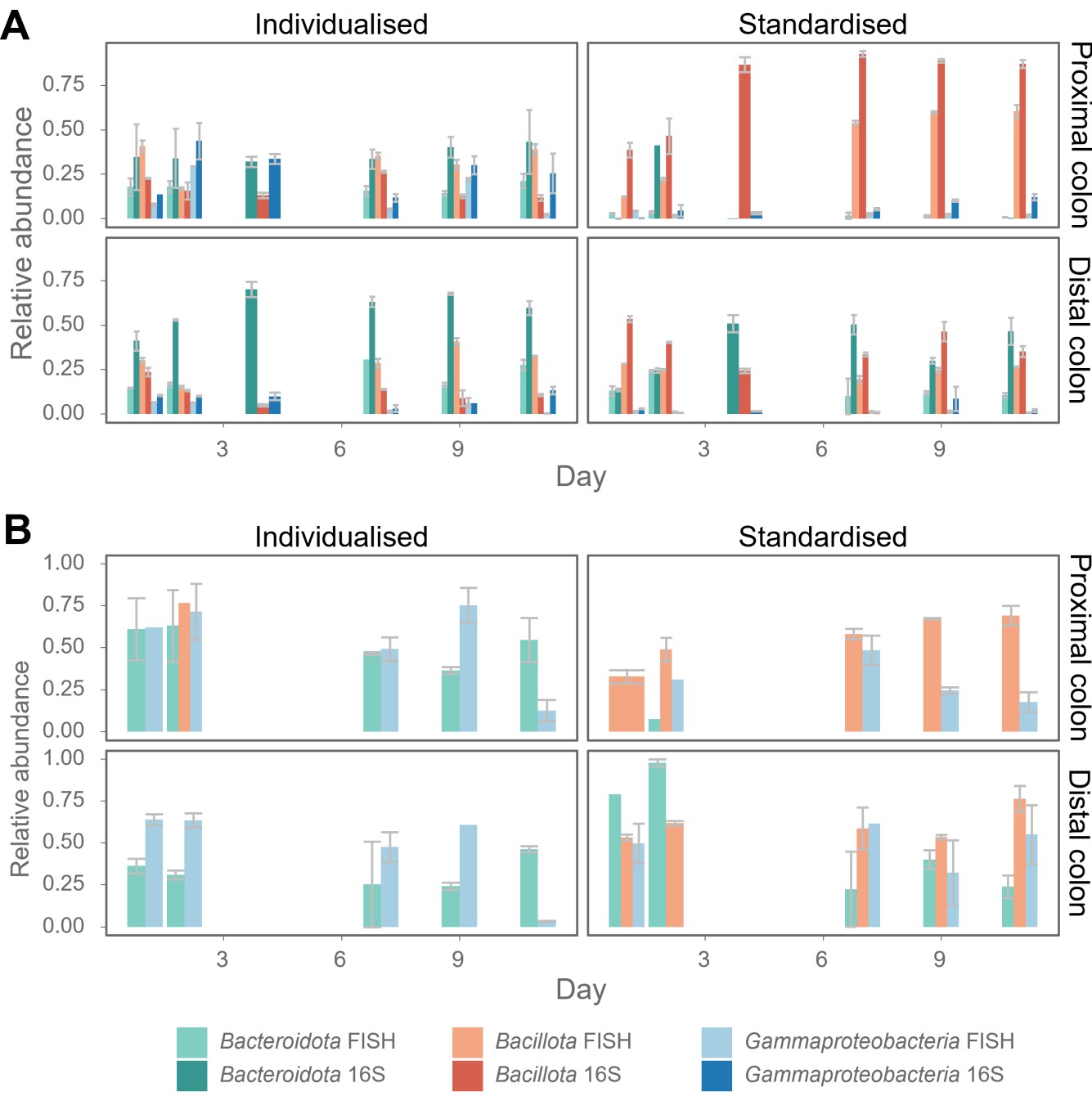

**FIG 6** (A) Comparison of 16S rRNA gene amplicon sequencing data (copy number corrected, identified reads) and FISH data of *in vitro* simulated gut samples. These samples were derived from a SHIME either individualized (with individualized feeding frequencies, media, and transit time based on each fecal donor, Supporting Information S1) or using the standardized protocol (with fixed feeding frequencies, media, and transit time, Supporting Information S1) (24). Biological replicates were averaged, and standard error is shown. In blue, two different taxonomic levels are displayed. The *Gammaproteobacteria* were targeted by FISH, while the sequencing abundance displays the higher classification rank of *Proteobacteria*. However, the sequencing data reveal that almost 100% of the *Proteobacteria* are classified as *Gammaproteobacteria*. (B) The ratio of FISH/16S rRNA gene amplicon sequencing relative abundances representing the potentially active fraction of the population. Values exceeding 1 were omitted.

cell), accessibility of the probe-binding site on the ribosome, and cell permeabilization (51–53). Our results (Fig. 4) confirm that the hybridization efficiency of the three EUB338 probes mixture, targeting all bacteria, was consistent during the experiment. On a lower taxonomic level, the number of hybridized cells was more variable. This is reasonable,

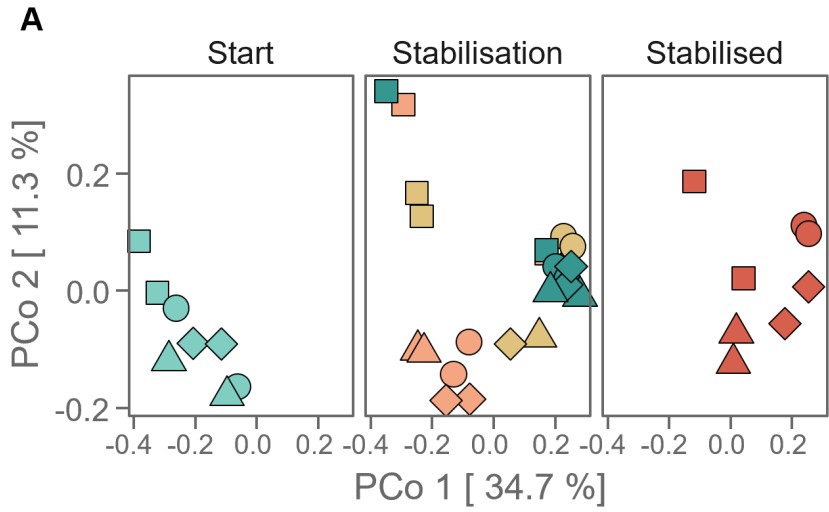

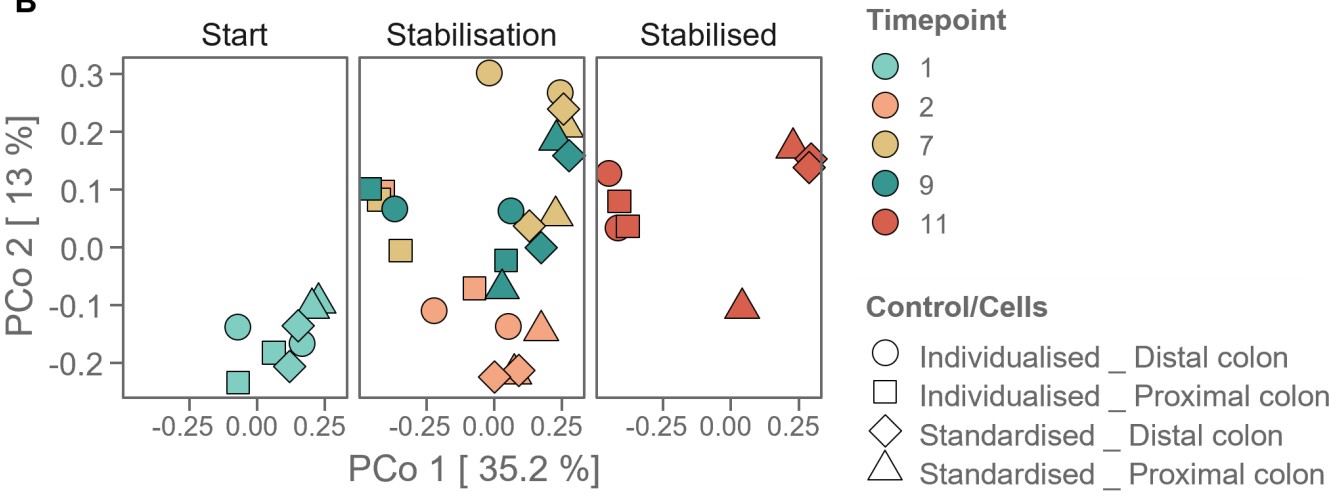

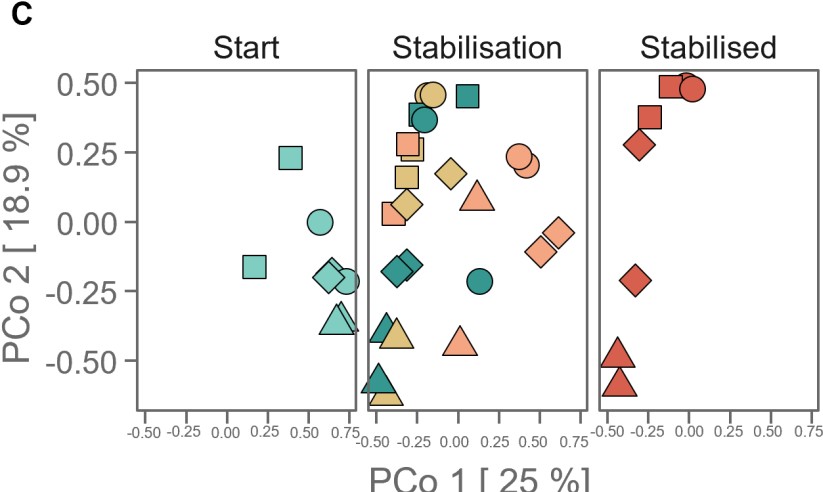

**FIG 7** PCoA diversity analysis of the simulated gut samples calculated on the FISH-stained cells of the taxonomic group *Bacillota* (Bary-Curtis dissimilarity matrix) by *Phenoflow* (4). The different vessels and conditions are indicated by different shapes, and different time points by color. The panels represent the different phases that occur during the start-up of a SHIME system (start—stabilization and stabilized). (A) Fingerprinting of all cells, including the FISH labeling; (B) Fingerprinting of the FISH-labeled *Bacillota* cells (C) Fingerprinting of the *Bacillota* classified ASVs.

TABLE 3 Qualitative comparison between the different fingerprinting techniques discussed and applied in this research; flow cytometry with general nucleic acid staining, flow cytometry combined with FISH and sequencing techniques

| | Flow cytometry with nucleic acid staining, without fixation | Flow cytometry with nucleic acid staining, with fixation | Flow cytometry—FISH | Amplicon sequencing techniques |
|---|---|---|---|---|
| Type of information | Absolute cell counts, phenotypic heterogeneity, intact/damaged[a] | Phenotypic heterogeneity | Phenotypic & taxonomic heterogeneity, ribosomal activity | Relative abundances, taxonomic heterogeneity |
| Target | Unspecific nucleic acids | Unspecific nucleic acids | Specific rRNA (+DNA counterstain) | Unspecific DNA (+eDNA) |
| Throughput time | Low (1 h) | Low (1 h + overnight fixation) | Medium (4 h + overnight fixation) | High (2–3 days) |
| Single cell | Yes | Yes | Yes | No |
| Consumable cost | Low (€2.5) | Low (€2.5) | Low (€5) | High (€20–200) |

[a]Depending on the type of staining used, some features of cells can be highlighted.

as it has been shown that rRNA concentrations can be variable during different growth stages of bacteria and can also differ significantly among taxa (22). The information obtained from FISH is very valuable as it allows us to investigate the potentially active fraction of a specific taxonomic group of cells to, for example, investigate phenotypic changes in the cells. Nonetheless, it is important to consider other confounding factors that influence the number of FISH-targeted cells. As multiple probes are used, the formamide concentration is adjusted to be optimal for the mixture of probes, which might deviate from the optimal concentration of a single probe. This could cause some nonspecific binding. By using control samples, the effects of the different hybridization concentrations can be estimated. However, when fingerprinting, the deviation from the optimal formamide concentration need not have consequences in the diversity analysis, as the same deviation is present in all samples.

The FISH labeling of the marine samples increased the distance (and effect size of the comparison) between the centroids of the two different subject groups and the correlation with the 16S rRNA gene amplicon sequencing distance matrix, but it did not have better predictive power for grouping variables. In summary, the diversity between the samples, belonging to a specific plastic type, is much more prominent for the FISH labeling. This allows, for example, to track and distinguish the microbial response to different types of plastics in the context of an enrichment. For the SHIME samples, FISH labeling increased the effect size of the comparison between the centroids of two different subject groups (PFA fixed), the correlation with the 16S rRNA gene amplicon sequencing distance matrix (PFA and ethanol fixed), had a better predictive power for grouping variables (PFA fixed) but did not increase the distance between the centroids (both fixation methods). The PFA-fixed samples, targeted by two probes, seem to have an additional advantage. This might be due to the more prominent presence/absence of *Bacteroidota* and *Gammaproteobacteria* in the individualized and standardized SHIME operation (Fig. 6). In this case, the increased predictive power allows us to classify unknown samples as individualized or standardized. In a more applied context within human gut research, a technique like this allows us to follow up the ratio of *Firmicutes* over *Bacteroidetes*, the phenotypic diversity within these taxonomic groups, and predict gut health (54). In conclusion, the FISH-stained measurements did show to be more informative than what could be obtained through classic FCM (i.e., ribosomal activity, taxonomic group abundances, potential of fingerprinting one specific taxonomic group). Although depending on the application, general fixation and nucleic acid staining can be sufficient.

The diversity analysis of the staining methods applied to the marine bacteria indicated that, while the 16S rRNA gene amplicon sequencing data could separate the samples based on plastic type, the variance in the non-fixed staining method was rather explained according to time in the PCoA analysis (Fig. 2B). When looking into the (fluorescence-based) scatterplots of the data, the (multivariate/bivariate) distribution of

the data (on which the fingerprint is based) for the non-fixed samples is very different for day 0, compared to the other timepoints (Fig. S1A and B). Furthermore, it can be observed that the non-fixed inoculum samples are much harder to discriminate from the background than the samples later in time (Fig. S1C). This can be due to the change in physiology when inoculating a reactor from an original batch culture. When the samples are fixed in PFA (and have thus undergone several washing steps), cells can be distinguished from the background without any issues from day 0 onward (Fig. 2C and D). Next to preserving cells, the PFA fixation increases the mechanical strength or stability of the tissue, and the ethanol introduced during the fixation allows cells to be permeabilized (55). Contradictorily, it has been reported that when using aldehydes or alcohols as fixatives, light scatter signals are distorted, and reliable separation of bacterial cell populations from cell debris and background noise of a flow cytometer is hindered (56). Moreover, loss of cells and loss of fluorescence have been reported because of fixation (57), although loss of fluorescence is not observed in the current study nor in the studies by Cichocki et al. (58) and Emerson et al. (59). These results thus showcase that fixation of samples can avoid bias in regular flow cytometry measurements. Therefore, fixation is recommended for fingerprinting purposes, especially when investigating time-series experiments and no information on viability is required.

When comparing the results of the two different ecosystems, it should be noted that the experimental groups of the marine bacteria on different plastics and the different stages and groups of the SHIME had different taxonomic dissimilarity to begin with. While the comparison between the different groups of the marine samples has a pseudo $F$-ratio of 43.24, the pseudo $F$-ratio in the SHIME samples comparison was only 10.34 for the 16S rRNA gene amplicon sequencing data. Larger $F$-ratios indicate more pronounced group separation, in combination with the significance of this ratio (60). This is visualized in the two-dimensional PCoA plot of the sequencing data, where there is no overlap when plotting the trickling filter diversity, while there is for the SHIME plot (Fig. 2A and 3A).

For the marine ecosystem samples, it can be observed that overall, the combination of fixation and SG staining performed worst (second lowest distance between centroids, lowest correlation with the sequencing data, and lowest performance in predictive modeling). In comparison, the non-fixed, SG-stained samples performed better for the diversity analysis and had the highest predictive power (Table S3). As such, the results from the PCoA analysis, showing only two dimensions of variation, can be misleading (Fig. 2). Through ordination, we were only able to discriminate the non-fixed SG samples of timepoint 0 from the later timepoints, while the multivariate data discriminated between the samples of different plastic types. Furthermore, comparing non-fixed and fixed SG staining methods, the lower performance of the latter might not be due to the dye. SG is furthermore reported to have similar sensitivity as DAPI for agarose gel electrophoresis (61), and even a higher sensitivity for fixed, marine samples measured by microscopy (62). When using DAPI staining, the centroid discrimination and pseudo $F$-ratio are increased compared to both SG staining methods, but the AUC for the predictions is lower (Table S3). In summary, it is not possible to deduce a strong advantage of fixation or type of nucleic acid dyes for diversity estimations for this data set. The SHIME samples, on the other hand, showed only a slight increase in pseudo $F$-ratio and in predictive modeling when staining with FISH probes for *Gammaproteobacteria* and *Bacteroidetes* (and fixed in PFA) compared to the other methods (except for the correlation with 16S rRNA gene amplicon sequencing data). SGPI makes it possible to distinguish between cells with a damaged and an intact membrane, which could have influenced the diversity analysis (adding functionality, compared to a general nucleic acid stain).

Despite the fact that combining FISH with flow cytometry is a more laborious method (than classic flow cytometry combined with nucleic acid staining), the improvements in fingerprinting, increased centroid discrimination (more distinct in the plastic degradation data set), the possibility to have an estimate of (potentially active) abundances of specific

taxonomic groups (Fig. 5 and 6), and the ability to fingerprint a specific taxonomic group throughout an experiment (as shown here for the *Bacillota* in the SHIME data set, Fig. 7), make it a very valuable method. In addition, it could be a good screening technique, able to estimate in high throughput the composition and diversity of samples, thereby helping to obtain information to make a data-driven decision on which samples you want to process for more expensive and time-consuming sequencing techniques. Moreover, this technique provides single-cell and absolute abundance information that cannot be obtained by conventional sequencing techniques. Compared to FCM fingerprinting with general nucleic acid staining, additional information on taxonomy and ribosomal activity can be obtained (probed cells versus total cells, comparison of probe fluorescence intensity across samples) (Table 3). Furthermore, as proven with the *Bacillota* subgroup fingerprinting (Fig. 7), taxonomically labeled subpopulations (by FISH) can be separately investigated on phenotypic heterogeneity, by fingerprinting one taxonomic group across different samples. This was also demonstrated for a genus-level FISH probe, targeting *Alcanivorax* (63). The advantage of targeting rRNA instead of total nucleic acid content (regular FCM and amplicon sequencing) is that it enables the discrimination of the microbial population of interest (i.e., the potentially active).

In conclusion, the proposed combination of flow cytometry and FISH does increase the obtained information from one single measurement markedly by combining phenotypic, taxonomic, and physiological (activity) information. This additional information could prove to be an advantage to regular flow cytometry with nucleic acid staining in different fields of microbiology.

## AUTHOR AFFILIATIONS

[1]Centre for Microbial Ecology and Technology (CMET), Universiteit Gent, Ghent, Belgium
[2]KYTOS, Ghent, Belgium
[3]Centre for Advanced Process Technology for Urban Resource Recovery (CAPTURE), Ghent, Belgium

## AUTHOR ORCIDs

Valérie Mattelin http://orcid.org/0000-0001-8202-5022
Josefien Van Landuyt http://orcid.org/0000-0003-1611-1525
Yorick Minnebo http://orcid.org/0000-0002-1980-7117
Nico Boon http://orcid.org/0000-0002-7734-3103

## FUNDING

| Funder | Grant(s) | Author(s) |
| --- | --- | --- |
| Fonds Wetenschappelijk Onderzoek | 1288224N | Josefien Van Landuyt |
| Agentschap Innoveren en Ondernemen | HBC.2019.2622 | Valérie Mattelin |
| Bijzonder Onderzoeksfonds UGent | BOF.BAS.2022.0014.01 | Nico Boon |
| Fonds Wetenschappelijk Onderzoek | 30770923 | Yorick Minnebo |

## AUTHOR CONTRIBUTIONS

Valérie Mattelin, Conceptualization, Data curation, Formal analysis, Funding acquisition, Investigation, Methodology, Validation, Visualization, Writing – original draft, Writing – review and editing | Josefien Van Landuyt, Conceptualization, Data curation, Formal analysis, Funding acquisition, Investigation, Methodology, Validation, Visualization, Writing – original draft, Writing – review and editing | Frederiek-Maarten Kerkhof, Methodology, Supervision, Writing – review and editing | Yorick Minnebo, Formal analysis, Investigation, Validation, Writing – review and editing | Nico Boon, Funding acquisition, Methodology, Project administration, Resources, Supervision, Writing – review and editing

## DATA AVAILABILITY

The raw 16S rRNA gene amplicon data (marine) for this study can be found in the National Centre for Biotechnology Information (NCBI) database in BioProject PRJNA878852 and European Bioinformatics Institute's (EBI) European Nucleotide Archive (ENA) (SHIME) with accession number ERP143206. Flow cytometry data (.fcs format) are available in the FlowRepository archive under repository ID FR-FCM-Z7ZR (marine) and FR-FCM-Z6Z6 (SHIME).

## ADDITIONAL FILES

The following material is available online.

### Supplemental Material

**Supplemental materials (Spectrum01973-24-s0001.docx).** Supplemental figures, text, and tables.

### Open Peer Review

**PEER REVIEW HISTORY (review-history.pdf).** An accounting of the reviewer comments and feedback.

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
