## [Reviewer comments · Microbiology Spectrum]

Microbiology Spectrum

Integrating Taxonomic and Phenotypic Information through FISH-enhanced Flow Cytometry for Microbial Community Dynamics Analysis

Valérie Mattelin, Josefien Van Landuyt, Frederiek-Maarten Kerkhof, Yorick Minnebo, and Nico Boon

Corresponding Author(s): Nico Boon, Universiteit Gent

Review Timeline:

Submission Date:	August 6, 2024
Editorial Decision:	December 9, 2024
Revision Received:	April 14, 2025
Accepted:	April 28, 2025

Editor: Adriana Lopes dos Santos

Reviewer(s): The reviewers have opted to remain anonymous.

Transaction Report:

DOI: <https://doi.org/10.1128/spectrum.01973-24>

Re: Spectrum01973-24 (Integrating Taxonomic and Phenotypic Information through FISH-enhanced Flow Cytometry for Microbial Community Dynamics Analysis)

Dear Prof. Nico Boon:

We appreciate your submission and the effort you have invested in your work. However, after reviewing the manuscript, we have noticed several areas where the clarity could be significantly improved. To ensure your research is accessible and impactful, we kindly request that you revise the paper to enhance its language and coherence. This includes refining the scope of the article and the extensively documented the phenotypic scrutiny proposed, as well as polishing grammar, sentence structure, and word choice to convey your results more effectively.

Below you will find my comments, instructions from the Spectrum editorial office, and the reviewer comments.

Revision Guidelines

Sincerely,
Adriana Lopes dos Santos
Editor
Microbiology Spectrum

Reviewer #2 (Comments for the Author):

Hello,

In their research article "Integrating Taxonomic and Phenotypic Information through FISH-enhanced Flow Cytometry for Microbial Community Dynamics Analysis", the authors compare the beta diversity patterns and relative abundances of selected taxa obtained using well-known DNA probing (FISH) and cell sorting technologies (FCM) to that obtained using 16S amplicon sequencing, for microbiome samples obtained after two types of incubation experiments, a bioreactor treated with the addition of two different types of plastic, and a synthetic human gut sampled along time. The authors subject microbial communities recovered after incubations to cell fixation and/or staining and FISH fluorescence labelling treatments that vary depending on the experiment, to assess whether observed diversity and relative abundances are the same to 16S as a measure of performance.

The authors research is absolutely relevant to the need for more research comparing methods, to improve speed and accuracy when describing environmental microbial communities and their response to change, and notably to develop more insightful methods accounting for phenotypic trait and functions. Yet, I struggle to find how these aspects are addressed in this work, and primarily how the "physical traits and functions" and phenotypes highlighted in the title, abstract and importance sections are measured or used in a innovative manner beyond what is already been done using FISH and flow cytometry.

FISH and flow cytometry have been used in combination for a very long time, as already reported in the Amann and Fuchs's review of 2008 (Nat Rev Microbiol), which is has been cited 962 times. One recent article by Hill an Papoutsakis (2024, <https://doi.org/10.1128/msystems.00572-24>) reveals mechanisms (e.g., cell fusion) and accounts for culture growth phases, which the authors of the present works invoke in the abstract as major phenotypic diversity confounders justifying a FISH-flow cytometry approach. Hence, the claim that "incorporating taxonomic information might increase the resolving power of the microbial fingerprint" is not a suggestion for improvement as this is already established. Moreover, the following statement (l.110) assumes that "Fluorescent in situ hybridisation (FISH) [is a] rRNA-based method", which is not true as other genes and mRNA can be labeled (e.g., <https://doi.org/10.7717/peerj.8806>). Such inaccuracies and all my comments that follows could easily be addressed by rephrasing sentences in a revised version that would necessarily refocus the scope. If the scope is to remain purely methodological, it would be valuable to highlight the novelty of the current work in a more systematic comparison to previous FISH-FCM applications.

Indeed, my main comment is that research scope presented in the abstract and objectives is not really what the results address. Accessing "phenotypic information" and "physical traits and functions" are the main goals and proposed improvements, but the results do not show any detail about phenotypes. Of course, FCM does rely on morphological features but which features, and their importance to determining the levels of accuracy after different incubation and fixation/labelling experiments is not documented. For example, Brüwer et al. 2023 (mSystems) shows FISH-FCM results on cell volume, ribosome content and cell division (frequency to predict rate) and performed such detailed scrutiny for four taxa (SAR11, SAR86, Bacteroidetes and Aurantivirga) for which the genomes were reconstructed and growth rates estimated, so that sequencing data could be used as a relevant, and thoroughly described ground truth to the FISH-FCM data. In fact, from the title of the article, one would expect more insights about the precise kind of phenotypic information that would be exploited in the analysis of microbiomes, whereas the dependence to 16S data is eluded. In a revision, the fact that the results consist of a comparison with 16S would need to be highlighted.

Since the whole research presented here is based on comparing FISH-FCM to 16S sequencing data, the presented 16S results should also be documented in sufficient detail for a comprehensive assessment of the FISH-FCM results. The authors acknowledge in the Introduction that "(omics) methods [are] prone to technical biases, such as extraction and PCR bias". Thus, the choice of 16S data as sole reference is questionable given biases inherent to this method. In fact, background information about the potential biases and notably at detecting the taxa for which probes have been designed, should be reviewed and provided more extensively.

I see no flaw with methods, experimental data, and results, except that some methodological details are not provided comprehensively enough within the paper, and notably about the 16S experiment. Besides, there is also a lack of info about the controlled experiments from which the results derive, which makes it difficult to understand the systematic 16S vs FISH-FCM comparisons. Indeed, two different experiments (bioreactor and SHIME) are performed after different fixation and labelling assays, which are not described in the workflow proposed in Fig 1. Thus, it is hard to follow between methods and results sections.

Abstract and Introduction

The abstract seems out of sync with the results and makes statements that are either unclear or important but undocumented (incl. methodological). I miss the single-cell results, as all results deal with community-level analysis or single taxa (genus levels). This is also strongly alluded to in the introduction (l.126): "Consequently, phenotypic fingerprinting could be performed per taxonomic group, gathering single-cell information of this specific group within a population." Thus, I suggest irrelevances are trimmed and replaced by info about the presented research, and critically to explain how the methodological assessment fully relies on comparisons to 16S fingerprinting as a reference, using which beta diversity metrics and which differences between which experiment treatments. The underlying assumptions about 16S as ground truth and about the experimentally-generated populations tested to evaluate the approaches abilities at discriminating treatments, must be introduced in the abstract as well as

the expected value of using "two completely different ecosystems were considered" for evaluating the accuracy of the method.

I.57: "The two examples show that the described technique is versatile": I assume these two examples refer to SHIME and bioreactor? If so, could high versatility refer to the ability to for FISH and FCM to run on samples derived from such experiments? if so, than this would not be a new demonstration as this methods have been applied to multiple ecosystems. Or, is versatility referring to the ability to detect differences between different fixation and staining treatments (as for the 16S ground truth)? If so, the results are in fact more nuanced as some treatments do not reveal differences. Hence, if versatility here is about using any sort of fixation or staining, then please be more accurate in reporting the finding of which protocol works best at recovering the patterns observed using 16S.

The sentence I.128 is not clear and tautological. "monitor the response of bacteria down to small process differences" is not clear: what sort of process? and what differences?; remove "thus adding more detail" as it is potentially and not clear (more detail can be more noise so more info about the kind of details and how useful these would be is necessary); "focussing on the taxonomic groups playing a major functional role" and the "monitor[ing] the response of bacteria down to small process differences" is .

This sentence is an hypothesis but what it states is very general and already verified as FISH-FCM has already known to perform well. A more precise hypothesis related to the work performed and that consists in a systematic comparison with 16S would need to the reformulated.

I.134: one objective was that "FISH probe efficiency was investigated", but I struggle to see a thorough report on the efficiencies of each of the 10 probes presented on Suppl table 1.

I.134: what is actually performed should be details instead of the more general "the microbial diversity and dynamics of the different ecosystems were calculated". Indeed, no alpha diversity indices was calculated or no analysis calculating statistics based on the time component was performed. In fact, only beta diversity differences using bray-curtis, relative abundance comparisons and permutation tests were done. This should be clear. Note that using relative abundances for compositional data such as 16S amplicon sequencing has shortcomings that would need to be solved or discussed since the 16S serves as ground truth: see Gloor et al. (2017 <https://doi.org/10.3389/fmicb.2017.02224>) and Quinn et al. (2019 <https://doi.org/10.1093/gigascience/giz107>)

Methods

The methods need revision for as non-experts in both FISH and flow cytometry that I believe could both be interested. Useful info for comprehension is missing, some methodological details need clarification, some concepts that are obvious to flow cytometry experts need to be demystified for non-experts, and to understand the whole procedure and goals, analytical choices must be justified with more accurate information from cited literature, which is particularly true for the chosen probes and the FCM analysis parameters.

The "Sampling" section could summarise the sampling units and terms associated with them (which samples are fixed how, and their sampling times) to more easily understand which methods connect to which results presented in the text and figures, as a way to present the complete workflow, as it is not true that (I.169) "The complete proposed workflow is shown in Fig. 1.". Please consider expanding Fig.1, or providing a table presenting the samples and their features in a systematic way e.g., with columns such as "origin" ("bioreactor effluent" or "synthetic gut"). In particular, this figure/table details should introduce the names of the "groups" classified by the random forest tests as what is attempted is unclear.

In this last machine learning section are referred the "different staining methods in both ecosystems" and so it would be good to clarify what are all these in the "Sampling" part as a reader must be able to grasp all experimental design aspects in the main manuscript. Hence, please move the supplement info about sampling into the main document.

I question the relevance of "marine" samples. First, marine ecosystems are highly heterogeneous and so it is not clear whether these are water, sediment or any other substrate, and thus, how were the probes selected for. Also, it seems that "marine ecosystem samples originated from the effluent of four trickling filter bioreactors". What would "originated" means? Are the samples bioreactor effluents discharged into and then sampled from the marine environment? Or, as I suspect, were these marine samples added into the trickling filters, and then the effluent of the reactors were sampled? Please revise according, e.g. "Four marine samples were added into trickling filter bioreactors: two containing the plastic poly(3-hydroxybutyrate-co-3-hydroxyhexanoate) (PHBH) and two containing a novel plastic of PI-protected chemical composition (hereafter B4PF01)". Could "P" and "F" reactors be defined here instead of being re-defined later again in the Results sections?

Because of the high amount of methods and reagents used, the reading is complicated by frequent sentence interruptions by in-parentheses versions, brand/catalogue info, wavelengths, acronyms and, my above concern, references to sample types and experimental treatment that are only described the missing "Sampling" methods section. While I applaud the level of details about the material used and steps involved in FISH and FCM, I suggest the author consider evacuating all these highly technical info into a STAR methods table (see <https://www.elsevier.com/researcher/author/tools-and-resources/key-resources-table>), with sections for the different aspects of the experiments (sampling), fixation, dyes, etc.

The 16S experiment is based on DNA extracts and not RNA (cDNA). Hence, it should be referred to as "16S rRNA gene" or "16S rDNA" to avoid confusion.

It is insufficient to state an "in-house phenol-chloroform DNA extraction protocol". More details or a reference describing the extraction procedure in details is needed to allow for review and potential replication.

What justifies using tSNE and NMDS in addition to PCoA?

Results

It is confusing that novel acronyms are introduced here. Please define these in the "Sampling" section of the methods so that there is no need to redefine treatment groups. These should be clear as early as possible as the most important here is for the reader to understand what are the different treatments that are compared and how these comparisons are diagnostic to measuring the improvement offered by the proposed methodological innovation. In fact, this first result paragraph is to be moved entirely to the "Sampling" methods (and potentially summarized as a table with the treatments, sample sizes etc?), before the more details about underlying fixation, hybridization and staining methodologies (which are less important and could also be summarized into a STAR table).

The results highlight a lack of details in the methods and further results are necessary to understand the reason for the existence of clusters in the 16S results used as reference for the FISH-FCM data, e.g., is there lower alpha diversity in some treatments? I could find no sample and sample-prep metadata information that would indicate the potential effects of extraction of sequencing batch. Also, it is said that negative and positive controls are used but there is not report about these.

It appears that no actual phenotypic information is being extracted from the multiple FCM experiments, challenging what is announced in the abstract. Is it true that data derived from flow cytometry relies on these features but research aiming to establish improvement of a methods relying on these (as per the title) would be expected to provide in-depth descriptions of the measures phenotypic variation, which is possible as shown in other papers demonstrating the method.

In fact, the quality of the FISH-FCM method is measured based on matching to the 16S data, by checking whether the same diversity patterns are obtained, so the assumption is that the 16S result is a good standard, which is not inline with how this is presented in the intro where 16S is explained as being more biased. Hence, one would expect that an improved FISH-FCM method would yield different results, incl. non-significant Mantel correlations. This calls for alternative ground truth data, which is not available here. One solution for ground truthing would be a compelling experiment with known microbial composition and known dynamics (like in Brüwer et al. 2023 mSystems). Since the data stems from experimental inoculates originating from complex environments, such ground truth could be achieved but then the experiments must be described in great detail in the main document and additional about the 16S read abundances and diversity should be presented. For the bioreactors, however, it appears that a plastic degradation process is being referred to as a driver of community dynamics, but this process is not introduced or described and thus fails to convince as having ground truth value.

The process of monitoring plastic biodegradation in the reactors is mentioned throughout the manuscript, incl. Abstract, Intro, and also in the Results, when it comes as a new goal (l.242 "to clarify the plastic degradation process"). However, this issue is not documented by any result, and in the Results section, it appears as a plain discussion element with references (l.334), which should be moved to the Discussion section, or removed altogether as the plastic aspect seems irrelevant. I recommend either expanding on the significance of the plastic degradation process with insightful data and results in relation to your fixation treatments and FISH-FCM findings, or removing it (incl. in suppl methods "set up to investigate plastic biodegradation (unpublished data)"). Alternatively, this could be moved to the methods to justify the probes choice instead of "FISH probes were chosen based on literature and Probas"?

Please stick to describing results in the "Results" section and avoid adding experimental reasoning and justifications at this point. For example, the fact that different plastics may be/are different carbon sources in the experiment should be introduced before if relevant in the results, either in the introduction. Whether communities separate because of the presence of plastics, and whether these plastics represent different carbon sources may be a discussion element, but certainly not fall in the midst of the results. Moreover, for this aspect to be discussed properly, it would be necessary to know the plastic dosage and the presence of other, less refractory carbon sources that are likely present in bioreactors. Then, more info about the reactors would be necessary (e.g., pre-treatment) as well as about the sample (sediment? water?).

Also, I would like to argue that "FISH staining" is inappropriate as FISH is not a staining method per se because it is based on the hybridization of fluorescent probes, in contrary to Sybr Green, propidium iodine and DAPI.

Importantly, I suggest that the cited research is cited in a more accurate and relevant manner in the Introduction statements. Here are two examples:

- l.80: "On the other hand, genotypic information has, up to now, been obtained mostly by more time-consuming and expensive (omics) methods, prone to technical biases, such as extraction and PCR bias, and without absolute microbial abundance

quantification (Ozel Duygan & van der Meer, 2022; Props et al., 2016a)." Ozel Duygan & van der Meer (2022) is about the potential and applications of machine learning to extract clusters and knowledge from FCM data alone, and only invokes omics as follow up tech, or in their introduction as powerful tools yet requiring complementary data such as FCM. This referenced work does not present actual results on costs, technical biases, or the ability of omics to infer microbial abundances. However, it is cited 7 times in the intro, incl. at places where it seems irrelevant (it is not dealing with "classical diversity analysis based on diversity indices like the Hill numbers")

- I.83: "However, more recently, big strides have been made to overcome PCR bias and time constraints, with for example the Oxford Nanopore Technology (ONT) (Zorz et al., 2023), where, in theory, (16S rRNA) gene sequencing of the community can be performed within a day's work without amplification". Here, Zorz et al. (2023) is about a portably ONT sequencing solution also relies on (full SSU) PCR, and thus would also be biased as for the standard Illumina 16S rRNA PCR protocol - which is what the comparison between these two tech actually show. In fact, I questions the very relevance of this sentence since the authors' work do not rely on long-range PCR or long-read sequencing.

Overall, I encourage the authors to read the cited works more carefully to only extract relevant results, that actually support statements, as it may be misleading for the readership.

Minor comments:

Citation are not formatted properly (redundant), e.g.

- "[...] described by Van Landuyt et al, (2020) (Van Landuyt et al., 2020)."

- "[...] according to Vandeputte et al, (2017) (Vandeputte et al., 2017)."

Or confusing:

- I.207: "Rubbens et al. (2020) (Rubbens et al., 2021c)"

- These citations are duplicated:

-- Rubbens, P., Props, R., Kerckhof, F.-M., Boon, N., & Waegeman, W. (2021a). Cytometric fingerprints of gut microbiota predict Crohn's disease state. *The ISME Journal*, 15(1), 354-358. <https://doi.org/10.1038/s41396-020-00762-4>

-- Favere, J., Buyschaert, B., Boon, N., & De Gussem, B. (2020a). Online microbial fingerprinting for quality management of drinking water: Full-scale event detection. *Water Research*, 170, 115353. <https://doi.org/10.1016/j.watres.2019.115353>

The citation to Van Landuyt et al. 2023 justifying the taxonomic specificity of FISH probes is not referenced properly and is not accessible. If the thesis can be made accessible, then it would also be good to provide context about how the experimental conditions in this thesis have demonstrated genus-level FISH probing relevant to the proposed methodological improvement, e.g., was there consistence across many samples? same PFA fixation? It is impossible to access and thus ascertain relevance.

I.44: "diversity analysis" (no hyphen)

I.96: "relatively large number of cells in a community": relative to what?

I.99: "(which can be done by different mathematical approaches)": either remove or describe some of the most useful approaches, to demystify such advances to an interested readership.

I.103: "(for example for determining the Crohn's disease state and monitoring drinking water quality (Favere et al., 2020a))": this citation of an accurate result could be a whole sentence without parentheses as such successful applications are relevant. However, this reference is about drinking water, not about determining the Crohn's disease state. Please add the reference about determining the Crohn's disease state: if this is "Rubbens et al., 2021a", please cite at the right place.

I.247: "Principal Coordinates Analysis (PCoA) was performed to diversify between microbial community fingerprints": it is not clear what " diversify" means in this context. Would you the author mean "distinguish"?

Figures:

- A comparison on Figure 4 is made across reactors, and so only then are introduced the reactors names and sample sizes. Please add this info on an expanded Fig1 clarifying the full experiment.

- I suggest that time points on the ordinations are colored using a gradient color scheme as there seem to exits interesting change dynamics in the data.

Supplement:

I.17: replace "The proximal and distal colon pH of 5.6-5.9, respectively, 6.6-6.9, was maintained with built-in pH controllers and pumps regulating the dosage of 0.5 M NaOH and HCl (Chem Lab, Zedelgem, Belgium)." by "The pH of proximal and distal colon vessels was kept within 5.6-5.9 and 6.6-6.9, respectively using controllers and buffer pumps (Chem Lab, Zedelgem, Belgium)." If pH ranges are given, is it relevant to indicate that there are "pH controllers and pumps regulating the dosage of 0.5

M NaOH and HCl"?

- The figure captions could remind which experiments one being looked at (SHIME vs bioreactor) and which treatment therein, even though this is evident from the legend, i.e., please elaborate figure too so the presented results refer to the rather complicated experimental design more clearly.

- Similarly, please add titles to the Suppl tables 3 and 4 for clarity: both start with Supplementary Table 3 - By means of three different statistical tests, the multivariate data was analysed. The Bray-Curtis dissimilarity matrix was used to test the distance between the centroids of the groups with different plastics (F and P)

Review

This review uses FlowFISH and cytometric fingerprinting in comparison to 16S amplicon sequencing analysis to describe dynamics of microbial communities in two different reactor setups.

It suggests that the cytometric information is similar although statistically less reliable as compared to the 16S analysis. It suggests that the variation/difference might be caused by physiological cell behavior which, for instance, allows to distinguish physiological states from each other, although the data coming from the reactor setups were (at least partly) coming only from adaptation phases in reactors. This needs clarification.

The topic is interesting. Also the biostatistics are good. However, the experiments themselves are not very well structured and not performed to the aim. There seems to be an indecision if the paper is a method paper and wants to test different staining and fixation methods or wants to compare the behavior of microbial communities in different (partly incomplete) reactor setups. Unnecessary or unclear experiments should be taken out. The paper itself is a bit wordy and should be shortened a lot and be written much more to the point, especially in the discussion. Also some papers are wrongly cited, which needs correction. Over-interpretation should be avoided, limitations should be discussed.

Specific remarks are listed below.

Introduction:

L93: Please be aware SybrGreen is a nucleic acid dye and does not measure only DNA. It does not give information on genome size, this can only be obtained by a DNA specific dye such as DAPI (there are abundant papers on this available).

L94: The Koch paper does not give any information on viability, this is not possible by fingerprinting. This paper invents the correlational analysis which allows to get information on possible functions of subsets of microorganisms. Please check also Props paper in this regard.

L100: if you are presenting working gate setting tools or microbial communities, please also take up the flowEMMi (2022)

Methods

L143: Fixation: This differentiates between gram positive and gram negative samples. As this study is on microbial communities, this method cannot be correct. Later in the paper it seems that always PFA fixation was used for all samples. Please check and change this part.

L155: Analar?

L169: What was the percentage of cells recovered from the filter?

L185: 'compromised' instead of 'deficient'

L211-2225: please see some spelling issues here, citations missing

L210-211: Only Cy5 was used for the red FI. Why using so many red channels here? What is the specific information of each of the channel?

Results:

L245: Did you also test a DAPI and/or a SYBRGreen staining of cells that were handled in a similar way as the FISH samples were handled but without probes? This better discrimination of cell types can also have been caused by the longer and more intensive handling of the cells using the FISH protocol.

Alternatively: e.g. for the DAPI you can take out all green and red discriminators and see, if the cell

handling might have had a discriminating effect. It would be interesting to see which information is coming from FISH probes and which from the fingerprint.

L252: I do not fully agree. To me it seems that this information is also given by DAPI staining and also by SG (fix) staining: the colors are not well chosen, differences can hardly be seen. Perhaps this can be improved. How does the PCoA shows if you would group timewise for each of the reactors?

L279-282: The Figures 2 and 3 show more samples, so I do not understand this sentence. Did you run different numbers of samples for the statistics? In this case which data (time points) did you use?

L287: citation missing?

L285: I do not find the difference between individualized and standardized protocols that were used here. Has this something to do with the fixation procedure? Or has this something to do with the feeding strategy? Please write the chapter in a much clearer way.

L298: In Figure 3: Maybe it would be good to show the 4 individual groups separately? It is also unclear to me what this Figure is highlighting: perhaps: The resolution between the fixation methods in both cases is not better compared to the 16S, so no additional (or no less) information could be reached by the FISH connected fingerprint. The SGPI does not work (as I would expect).

L297: Although Table SI 4 gives these types of numbers, the Figure 3 does not show this. For one, we have 4 groups. Second: SGPI is overlapping completely while A, C, and D are only partly overlapping. If homoscedasticity is referring to the degree of overlapping between the 4 groups I would expect a different result. If not, please explain or run the data anew.

L322: there was only 1 other staining method: SGPI, which did not work.

L323: Is this the headline of the following chapters? Please clarify the sections in a better way for better understanding

L333: Here is the question again: how many cells were already lost in the first filtration step?

L351-355: How can gram positive and gram negative cell from a community be differentiated before fixation? Please clarify.

Figure 6: the colors do not differentiate between Bacteroidota and Gammaproteobacteria. It also seems to me that the Bacillota FISH shows higher abundance compared to 16S. Please clarify. I also do not see a stabilized system in the different approaches, only the composition of the distal colon (standardized by what?) shows all three classes for 4 days. Please explain also here.

L364: Why was the β -diversity studied only for the Bacillota class? How many genera were involved? Or: did you differentiate between the 3 classes?

There is no Figure 8.

L372: If I understand correctly: the reactors were run for 11 days. The last day is declared as a stable situation. That there is a stable system was not shown. Day 1 is inoculum and Days 2 to 9 are the adaptation phase. Which dilution rate did you use? Was this the same for all reactors? If not, I'm unsure if these data can be compared.

I also do not agree to the separation of standardized-proximal colon samples from the other differentiation techniques. In A they are separated as well as in C but not in B (FISH labelled). Also true for the other samples. Please clarify.

Discussion

L378-381: this is very general and was not really shown in the paper.

L387 and following: If I understood correctly: only less than 50% of cells were labeled by the EUB probes. All was compared on the class level. Other data were not shown.

L392-398: such data were not shown. There are many publications that tested best FlowFISH conditions, maybe those can be cited here.

L416-417: I disagree, this was not found.

Section 218 to 436: the fingerprint of the fresh SG cells shows a lot of noise. For the differentiation: where the settings of the cytometer always the same? If cell density is low more noise is eminent in the 2D fingerprint, which set it also apart from the other samples. The Fresh SG from the microbiome did not show any differentiation between samples. I would be careful not to over-interpret these types of samples.

L428: PFA does not permeabilize cells. The protein structures of the surface are glued to each other which makes the cells more stable. The permeabilization is done by the ethanol.

L429-432: This is a wrong citation: this paper did not investigate PFA or ethanol fixations. The sentence comes from the introduction where other work was cited but which was not investigated in the paper. Please correct.

L432-433: the Cichocki paper used PFA ethanol fixation very successfully. But it did not investigate cell loss. Therefore this connection was also wrongly cited. It is very clear to the cited group that there is cell loss during fixations and staining steps when centrifugations steps are involved. Please do not mix up findings. Please correct.

L454: Surely this citation is not relevant for SG vs. DAPI staining.

L463: No, SGPI does not do this. PI can be taken up also by growing cells when the cell wall is synthesized. There are a bunch of publications on this. When you work in a community you never know which cell type is stained and why. Sort PI stained cells on agar plates: you will be surprised. The best would be to take this sample out of the whole study because it is meaningless.

474: as before, I think abundance information you can only get with fresh SG cell measurement, not with flowFish. I also do not see additional information compared to 16S with exception of Figure 2D (which is without FISH), Please be more precise and clarify.

ASM Spectrum

Ghent, 09/04/2025

Dear editor and reviewers,

We would like to thank you again for the constructive and valuable revisions of our manuscript.

We have addressed all comments and adjusted the manuscript accordingly. Point-by-point answers to all other feedback are provided below and include the revised sections of the manuscript (in blue) as well as the corresponding line numbers in the resubmitted manuscript (without tracked changes).

We hope these changes are satisfactory to you and we look forward to the further steps for our manuscript. We have added this letter to the submission as well.

Nico Boon on behalf of all authors,

Reviewer 1

This review uses FlowFISH and cytometric fingerprinting in comparison to 16S amplicon sequencing analysis to describe dynamics of microbial communities in two different reactor setups.

It suggest that the cytometric information is similar although statistically less reliable as compared to the 16S analysis. It suggest, that the variation/difference might be caused by physiological cell behavior which, for instance allows to distinguish physiological states from each other, although the data coming from the reactors set ups were (at least partly) coming only from adaptation phases in reactors. This needs clarification.

Thank you for your observation, clarifications have been added to the sampling part of the material and methods and throughout the manuscript.

The topic is interesting. Also the biostatistics are good. However, the experiments themselves are not very well structured and not performed to the aim. There seem to be an indecision if the paper is a method paper and wants to test different staining and fixation methods or wants to compare the behavior of microbial communities in different (partly incomplete) reactor setups. Unnecessary or unclear experiments should be taken out. The paper itself is a bit wordy and should be shortened a lot and be written much more to the point, especially in the discussion. Also some papers are wrongly cited, which needs correction. Over-interpretation should be avoided, limitations should be discussed.

Specific remarks are listed below.

Introduction:

- **L93: Please be aware SybrGreen is a nucleic acid dye and does not measure only DNA. It does not give information on genome size, this can only be obtained by a DNA specific dye such as DAPI (there are abundant papers on this available).**

We see your concern, that is why we specify we are talking about high nucleic acid content and low nucleic acid content, not only genome size (line 48-49).

- **L94: The Koch paper does not give any information on viability, this is not possible by fingerprinting. This paper invents the correlational analysis which allows to get information on possible functions of subsets of microorganisms. Please check also Props paper in this regard.**

Although a well-placed concern from you, this is why we have split the references so they address the right part of the sentence, with the Koch reference referring to HNA and LNA populations and the Props reference referring to cell viability.

- **L100: if you are presenting working gate setting tools or microbial communities, please also take up the flowEMMi (2022)**

A very interesting tool and nice suggestion from your part, we have added in the reference as you requested.

Methods

- **L143: Fixation: This differentiates between gram positive and gram negative samples. As this study is on microbial communities, this method cannot be correct. Later in the paper it seems that always PFA fixation was used for all samples. Please check and change this part.**

As you know, gram negative cells' outer membrane layer can contain lipopolysaccharides and they have a thinner cell wall which can impact the permeability and the stiffness of the cells, while gram positive bacteria have a thick cell wall intercalated with teichoic acids but no outer cell membrane. This wildly different permeability means you might need different fixation methods when fixing and permeabilizing the cells for FISH. We knew to expect more gram negative bacteria in the marine samples and more gram positive (or at least a more balanced mix) in the human gut type samples. Because of this, we decided to fix the marine samples with PFA (as is the standard for environmental samples with mainly gram negative bacteria), while testing out both fixation methods (ethanol & PFA) in the human gut samples (ethanol is the most common fixation method for gram positive bacteria).

- **L155: Analar?**

Analar is a commercial term used to describe analytical quality high purity chemicals usually 99.99% purity or better. Has been removed to avoid confusion.

- **L169: What was the percentage of cells recovered from the filter?**

We have not tested this as the results are used for relative calculations and not absolute.

- **L185: 'compromised' instead of 'deficient'**

We have adjusted this.

- **L211-2225: please see some spelling issues here, citations missing**

We thank you for your observance, we have adjusted spelling mistakes where we found some.

- **L210-211: Only Cy5 was used for the red FI. Why using so many red channels here? What is the specific information of each of the channel?**

Although Cy5 has a peak at 665nm, it has a quite long emission tail that reaches 800nm. We find it relevant to try and capture all the fluorescence emitted by the probes and thus decided

to use both the R660 channel and R783 channel. The red channel of the blue laser (B700) was taken into account as we used the fluorescent tag Atto490ls for the EUB probes which has an emission peak at 660nm. In case of the fresh samples, this channel is also relevant for the PI emission peak (where the PI is excited by the SG emission peak). Moreover, SG in itself has a very long emission peak as well, so it does cause some signal in the B700 as well. The use of multiple channels, including the optimal Cy5 (R660) and suboptimal (R783) channel, allows us to optimize the signal-to-noise ratio and obtain more comprehensive and precise measurements.

Results:

- **L245: Did you also test a DAPI and/or a SYBRGreen staining of cells that were handled in a similar way as the FISH samples were handled but without probes? This better discrimination of cell types can also have been caused by the longer and more intensive handling of the cells using the FISH protocol. Alternatively: e.g. for the DAPI you can take out all green and red discriminators and see, if the cell handling might have had a discriminating effect. It would be interesting to see which information is coming from FISH probes and which from the fingerprint.**

While there is some effect of dehydrating the cells when submitting them to the FISH protocol, we did not test whether this cell handling had a significant effect on the cells. However, we did perform the staining on the fixed samples, which is the most significant step in permeabilizing the cells (as the cells are kept in the freezer afterwards in 50% ethanol solutions) and we did test if the models gave similar results when not taking into account as many channels (in effect disregarding the fluorescence of the probes). However, this last approach did not yield as good results, it seemed to compromise the quality and depth of the data.

- **L252: I do not fully agree. To me it seems that this information is also given by DAPI staining and also by SG (fix) staining: the colors are not well chosen, differences can hardly be seen. Perhaps this can be improved. How does the PCoA shows if you would group timewise for each of the reactors?**

The colors, to us quite distinct colors, show the timewise grouping, which means fresh flow cytometry seems to be mostly grouped timewise (i.e. T0 is very different from all the other timepoints suggesting a high influence of growth stage over taxonomy). We have adjusted the figure so the hollow dots are now triangles to make it more clear, however, we do not agree that SG/DAPI gives the same information as the FISH – DAPI stained samples.

- **L279-282: The Figures2 and 3 show more samples, so I do not understand this sentence. Did you run different numbers of samples for the statistics? In this case which data (time points) did you use?**

For a correct comparison only the samples for which in parallel 16S rRNA gene amplicon sequencing was performed were used.

- **L287: citation missing?**

The reference was made to a phd thesis (Minnebo, 2023), we adjusted it to make it clearer

- **L285: I do not find the difference between individualized and standardized protocols that were used here. Has this something to do with the fixation procedure? Or has this something to do with the feeding strategy? Please write the chapter in a much clearer way.**

Thank you for your feedback. We had included a detailed explanation of the setup and sampling strategy in the supplementary information in the original manuscript. However, we understand the need for further clarification within the main manuscript. Therefore, we have now added a concise description of the setups in the materials and methods section and clarified these points within the results section for better clarity.

“Sample collection and experimental setups of both ecosystems are described in supporting information S1. In short, marine ecosystem samples were collected from the effluent of four trickling filter bioreactors: two with PHBH plastic (referred to as P) and two with a novel plastic, B4PF01 (referred to as F), to investigate plastic biodegradation. Effluent samples for fixation and flow cytometry were collected thrice weekly for 56 days. Samples for 16S rRNA gene amplicon sequencing were collected on days 0, 14, 28, 35, 42, and 56. In vitro simulated gut microbiome samples were derived from the Simulator of the Human Intestinal Microbial Ecosystem (SHIME, Prodigest, Zwijnaarde, Belgium). The SHIME model simulated the proximal and distal colon in different, but consecutive, vessels, with controlled pH, residence time, temperature (37°C), mixing (200 rpm), and diet (Minnebo et al., 2021; Molly et al., 1993; Van de Wiele et al., 2015). Two setups were used, each in duplicate. The first was a standardized SHIME with fixed eating patterns, transit time and nutritional media. The second was an individualized SHIME with adjusted parameters, according to faecal donor characteristics. Faecal donor samples were inoculated in the colon vessels, and the system stabilized over 11 days. Samples were collected on days 1, 2, 4, 7, 9, and 11 for analysis (FCM and 16S rRNA gene amplicon sequencing) (Minnebo, 2023).”

“The in vitro simulated gut microbiome in SHIME reactors were set up either individualised (with individualised feeding frequencies, media and transit time based on each faecal donor Supporting Information S1) or using the standardised protocol (with fixed feeding frequencies, media and transit time Supporting Information S1), to determine the importance of individualisation for future reference (Minnebo, 2023).”

- **L298: In Figure 3: Maybe it would be good to show the 4 individual groups separately? It is also unclear to me what this Figure is highlighting: perhaps: The resolution between the fixation methods in both cases is not better compared to the 16S, so no additional (or no less) information could be reached by the FISH connected fingerprint. The SGPI does not work (as I would expect).**

Thank you for your feedback. We understand that this figure may be overwhelming. However, we believe that displaying the four individual groups in the same plot adds significant value by highlighting the differences and similarities between the groups. We hope that by ensuring all axes, legends, and labels are clearly marked that it is easy to understand.

Additionally, the figure shows the differences between dissimilarity matrices calculated based on 16S rRNA gene amplicon sequencing, SGPI staining, DAPI and FISH staining of EtOH-fixed cells, and DAPI and FISH staining of PFA-fixed cells. Although less apparent than for the marine samples, there does appear to be an improved separation between individualized and standardized when applying FISH visually.

- **L297: Although Table SI 4 gives these types of numbers, the Figures 3 does not show this. For one, we have 4 groups. Second: SGPI is overlapping completely while A, C, and D are only partly overlapping. If homoscedasticity is referring to the degree of overlapping between the 4 groups I would expect a different result. If not, please explain or run the data anew.**

The analysis we performed is not only based on the two most discriminating axis, but is a multivariate analysis taking into account more than two dimensions and is thus a more accurate analysis than a visualization of only two dimensions, this is why there might be some discrepancy.

- **L322: there was only 1 other staining method: SGPI, which did not work.**

We respectfully don't know what the reviewer means by this comment, it is not stated in the text that we are comparing with more than one staining method. The staining of the SGPI worked (live (non-membrane compromised) cells were stained only with SG and damaged cells were stained with both SG and PI), the differentiation between the two systems based on the SGPI phenotypic fingerprint did not work well, as we state.

- **L323: Is this the headline of the following chapters? Please clarify the sections in a better way for better understanding**

Thank you for your feedback regarding the section headlines. We agree that clearer headlines are essential for better understanding. We have revised the headlines throughout the entire manuscript to improve clarity and readability:

Chapter

Subchapter

subsubchapter

- **L333: Here is the question again: how many cells were already lost in the first filtration step?**

We have not tested this as the results are used for relative calculations and not absolute. By keeping the filtration conditions equal for all samples, we ensure that any potential cell loss is uniformly distributed, thereby not affecting the relative comparisons we aim to make. However, this could be quantified in the future, and might be quite close to the original numbers. I hope this clarifies our approach. Please let us know if you have any further questions or require additional information.

- **L351-355: How can gram positive and gram negative cell from a community be differentiated before fixation? Please clarify.**

Thank you for your remark, we did not differentiate between gram positive and gram negative cells in the community before fixation, we just used two different fixation methods, one being the better one for gram positive and the other for gram negative ones, to see which one gives the best results when in doubt (like in the case of gut samples, where you have a big population of gram positive bacteria). We adjusted the text and hope that clarifies it for you.

“To avoid fixation problems and biases and to identify the best strategy in case of a community with mostly Gram positive bacteria, the SHIME samples were fixed in two different ways, with both ethanol and PFA. Most *Bacillota* are Gram positive, except the *Veillonella* genus (50 Amplicon Sequence Variants (ASVs) were classified as *Veillonella*), and require EtOH fixation, while *Bacteroidetes* and *Proteobacteria* are Gram negative bacteria, and are preferably fixed in PFA (Vesth et al., 2013; Wexler, 2007).”

- **Figure 6: the colors do not differentiate between Bacteroidota and Gammaproteobacteria. It also seems to me that the Bacillota FISH shows higher abundance compared to 16S. Please clarify. I also do not see a stabilized system in the different approaches, only the composition of the distal colon (standardized by what?) shows all three classes for 4 days. Please explain also here.**

Response: We understand your concern. We have added more clarity on the individualised and standardised nature of the setup in the caption:

“(A) Comparison of 16S rRNA gene sequencing data (copy number corrected, identified reads) and fluorescence in situ hybridisation (FISH) data of in vitro simulated gut samples. These samples were derived from a SHIME either individualised (with individualised feeding frequencies, media, and transit time based on each faecal donor, Supporting Information S1) or using the standardised protocol (with fixed feeding frequencies, media, and transit time, Supporting Information S1) (Minnebo, 2023). Biological replicates were averaged and standard error is shown. In blue, two different taxonomic levels are displayed. The Gammaproteobacteria were targeted by FISH, while the sequencing abundance displays the higher classification rank of Proteobacteria. However, the sequencing data reveals that almost 100% of the Proteobacteria are classified as Gammaproteobacteria. (B) The ratio of FISH/16S rRNA amplicon sequencing relative abundances representing the potentially active fraction of the population. Values exceeding 1 were omitted.”

- **L364: Why was the β -diversity studied only for the Bacillota class? How many genera were involved? Or: did you differentiate between the 3 classes?**

As the Bacillota class seemed to be most impacted by the different runs (i.e. a standardized run vs. an individualized run) based on the data we gathered, it seemed interesting to see if we could perform the analysis usually performed at total cell community level, at class level, and get interesting information out of it. This exercise could be performed for the other classes as well, if need be, but to us, this seemed the most interesting one.

- **There is no Figure 8.**

Thank you for pointing out the incorrect figure reference. We have corrected this error and now refer to Fig 7C as intended.

- **L372: If I understand correctly: the reactors were run for 11 days. The last day is declared as a stable situation. That there is a stable system was not shown. Day 1 is inoculum and Days 2 to 9 are the adaptation phase. Which dilution rate did you use? Was this the same for all reactors? If not, I'm unsure if these data can be compared.**

We have added more explanation of the stabilization in our SHIME system. Day 1 is indeed the inoculation day, and days 2 to 9 represent the adaptation phase. The system was stabilized by day 11, as shown in the added supplementary figure. The dilution rate was consistent throughout the run of the reactors:

”In our SHIME system, different phases were distinguished, with day 1 being the starting phase, day 2 – 9 as the stabilisation phase and day 11 as the stabilised phase, in which no fluctuations were noted in the for 2 consecutive timepoints (as shown in supplementary Figure 7). In the start phase, all different applied conditions...”

- **I also do not agree to the separation of standardized-proximal colon samples from the other differentiation techniques. In A they are separated as well as in C but not in B (FISH labelled). Also true for the other samples. Please clarify.**

We apologise for any confusion caused by the original sentence. Our intention was to convey that FISH allowed us to distinguish between individualised and standardised SHIME systems. To clarify this point, we have revised the manuscript accordingly:

“By day 11, in the stabilised system, the standardised-proximal colon condition can be clearly separated from the other conditions for the 16S rRNA gene amplicon sequencing data. **For the Bacillota (FISH) labelled cells, a clear separation between individualised and standardised SHIME conditions is seen (Figure 7B).**”

Discussion

- **L378-381: this is very general and was not really shown in the paper.**

We appreciate your feedback and have revised our manuscript to more accurately reflect our work:

“**In our study, we integrated phenotypic fingerprinting, a technique well-established across various ecosystems, (De Roy et al., 2012; Props et al., 2016a) with FISH, This combination allowed us to merge physiological, phenotypic, and taxonomic information, providing a more detailed description of the microbial community.** Moreover, the implementation of FISH provides information on the ribosomal content (which is a proxy for activity) within the microbial community. This study demonstrates that the implementation of FISH in flow cytometry for microbial diversity analysis provides more information on a single cell level to answer specific scientific questions.”

- **L387 and following: If I understood correctly: only less than 50% of cells were labeled by the EUB probes. All was compared on the class level. Other data were not shown.**

In this part of the paper we want to highlight the efficiency of FISH labelling to the readers. This, to us, is important information for critical interpretation of the results. As we aim to use higher-level taxonomic groups, only this data was presented.

- **L392-398: such data were not shown. There are many publications that tested best FlowFISH conditions, maybe those can be cited here.**

We are unsure to what the reviewer is referring here.

- **L416-417: I disagree, this was not found.**

Thank you for your feedback. We understand your concern regarding these statements. However, our FISH probes did reveal clear differences between standardised and individualised configurations, as shown in Figure 7B.

- **Section 218 to 436: the fingerprint of the fresh SG cells shows a lot of noise. For the differentiation: were the settings of the cytometer always the same? If cell density is low more noise is eminent in the 2D fingerprint, which set it also apart from the other samples. The Fresh SG from the microbiome did not show any differentiation between samples. I would be careful not to over-interpret these types of samples.**

Thank you for your concern, throughout the sampling campaign, fresh samples were run on the flow cytometer with the same settings (as described in the M&M) as it would not be possible to do phenotypic fingerprinting and direct comparison on samples that were not run on the same machine with the same settings. However, cell density was a bit lower in the starting condition than over time, which could have indeed caused a bit more noise within the cell gate, this is also the reason why we perform the phenotypic fingerprinting on the events identified as cells rather than all the measured events. Moreover, studies (like Garcia-Timmermans et al. 2020) have shown that it is possible that the physiological state of the cells (i.e. here start would mean lag phase (diluted culture for inoculation), versus later on (more stationary cells)) can have an effect on the fingerprint.

- **L428: PFA does not permeabilize cells. The protein structures of the surface are glued to each other which makes the cells more stable. The permeabilization is done by the ethanol.**

Thank you for pointing this out. We have adapted our manuscript:

“Next to preserving cells, PFA fixation increases the mechanical strength or stability of the tissue and **the ethanol introduced during the fixation** allows cells to be permeabilized”

- **L429-432: This is a wrong citation: this paper did not investigate PFA or ethanol fixations. The sentence comes from the introduction where other work was cited but which was not investigated in the paper. Please correct.**

Thank you for noticing, we have adjusted the citation to cite the work that was cited in the reference.

- **L432-433: the Cichocki paper used PFA ethanol fixation very successfully. But it did not investigate cell loss. Therefore this connection was also wrongly cited. It is very clear to the cited group that there is cell loss during fixations and staining steps when centrifugations steps are involved. Please do not mix up findings. Please correct.**

You are correct in the fact that there will always be some cell loss, we apologize if it was not clear that we were talking about the loss of fluorescence (that some other groups have noticed). We have adjusted the text to make it clearer.

“Moreover, loss of cells and loss of fluorescence have been reported because of fixation (Kamiya et al., 2007), **although loss of fluorescence** is not observed in the current study nor in the studies by XX et al () and YY et al ().”

- **L454: Surely this citation is not relevant for SG vs. DAPI staining.**

Although the work by Bourzac et al (2003) is based on pure DNA staining, the information we take from this paper, i.e. that the intensity coming from pure NA stained with either SG or DAPI is similar and higher than that of ethidium bromide does seem relevant to us. Of course we acknowledge that staining cells is not the same as staining pure NA, but when permeabilized by ethanol, the method of entrance of the stain and its efficiency becomes less relevant. Moreover, the research cited by Shibata et al (2007) discussing visualizing the density of the bacterial communities better with SG as with DAPI also seems relevant in the context of comparing DAPI staining of bacterial cells with SG staining of bacterial cells.

- **L463: No, SGPI does not do this. Pi can be taken up also by growing cells when the cell wall is synthesized. There are a bunch of publications on this. When you work in a community you never know which cell type is stained and why. Sort PI stained cells on agar plates: you will be surprised. The best would be to take this sample out of the whole study because it is meaningless.**

We respectfully disagree, it is widely accepted in the scientific community that PI is in general a membrane-impermeable fluorescent DNA stain ever since the method was established in 1999 by Boulos *et al.* (<https://www.sciencedirect.com/science/article/pii/S0167701299000482>) and commercialized by BacLight in 2004 (<https://onlinelibrary.wiley.com/doi/full/10.1002/cyto.a.20069>), meaning that it stains membrane-deficient cells. We however do agree with you that not all membrane-

damaged cells are non-viable (and some PI positive cells will be able to recover from their membrane damage and grow on agar plates when sorted), which is why we explicitly state “damaged” cells, rather than “dead” or “non-viable” cells. Moreover, when looking at biofilms, there might be a lot of eDNA within the EPS layer, giving a false positive PI result as discussed by Rosenberg *et al.* (2019) (<https://www.nature.com/articles/s41598-019-42906-3#Sec1>) and some papers suggest that high membrane potential, certain physiological processes, cell clumping and VBNC could affect the accuracy of PI staining (references within Rosenberg et al. 2019). However, for a general planktonic population, it is a widely accepted and suggested method to differentiate between (mostly) membrane-damaged and live cells and our own controls (*i.e.* heat-killed populations, freshly grown pure cultures, fresh inocula from different environmental communities and different populations within our group) confirm this.

To us, the SHIME example is relevant as it shows the technique can be used on completely different environmental microbial communities.

- **474: as before, I think abundance information you can only get with fresh SG cell measurement, not with flowFish. I also do not see additional information compared to 16S with exception of Figure 2D (which is without FISH), Please be more precise and clarify.**

To us it seems that this technique offers phenotypic information (*i.e.* from the general scatter channels like FSC and SSC as well as NA content (DAPI/SG) and to some extent cell numbers (although some cell loss will be inevitable it is uniform over all samples and can be as such quantified) as well as taxonomic level information.

Reviewer 2:

Hello,

In their research article "Integrating Taxonomic and Phenotypic Information through FISH-enhanced Flow Cytometry for Microbial Community Dynamics Analysis", the authors compare the beta diversity patterns and relative abundances of selected taxa obtained using well-known DNA probing (FISH) and cell sorting technologies (FCM) to that obtained using 16S amplicon sequencing, for microbiome samples obtained after two types of incubation experiments, a bioreactor treated with the addition of two different types of plastic, and a synthetic human gut sampled along time. The authors subject microbial communities recovered after incubations to cell fixation and/or staining and FISH fluorescence labelling treatments that vary depending on the experiment, to assess whether observed diversity and relative abundances are the same to 16S as a measure of performance.

The authors research is absolutely relevant to the need for more research comparing methods, to improve speed and accuracy when describing environmental microbial communities and their response to change, and notably to develop more insightful methods accounting for phenotypic trait and functions. Yet, I struggle to find how these aspects are addressed in this work, and primarily how the "physical traits and functions" and phenotypes highlighted in the title, abstract and importance sections are measured or used in a innovative manner beyond what is already been done using FISH and flow cytometry.

FISH and flow cytometry have been used in combination for a very long time, as already reported in the Amann and Fuchs's review of 2008 (Nat Rev Microbiol), which is has been cited 962 times. One recent article by Hill an Papoutsakis (2024, <https://doi.org/10.1128/msystems.00572-24>) reveals mechanisms (e.g., cell fusion) and accounts for culture growth phases, which the authors of the present works invoke in the abstract as major phenotypic diversity confounders justifying a FISH-flow cytometry approach. Hence, the claim that "incorporating taxonomic information might increase the resolving power of the microbial fingerprint" is not a suggestion for improvement as this is already established. Moreover, the following statement (l.110) assumes that "Fluorescent in situ hybridisation (FISH) [is a] a rRNA-based method", which is not true as other genes and mRNA can be labeled (e.g., <https://doi.org/10.7717/peerj.8806>). Such inaccuracies and all my comments that follows could easily be addressed by rephrasing sentences in a revised version that would necessarily refocus the scope. If the scope is to remain purely methodological, it would be valuable to highlight the novelty of the current work in a more systematic comparison to previous FISH-FCM applications.

Indeed, my main comment is that research scope presented in the abstract and objectives is

not really what the results address. Accessing "phenotypic information" and "physical traits and functions" are the main goals and proposed improvements, but the results do not show any detail about phenotypes. Of course, FCM does rely on morphological features but which features, and their importance to determining the levels of accuracy after different incubation and fixation/labelling experiments is not documented. For example, Brüwer et al. 2023 (mSystems) shows FISH-FCM results on cell volume, ribosome content and cell division (frequency to predict rate) and performed such detailed scrutiny for four taxa (SAR11, SAR86, Bacteroidetes and Aurantivirga) for which the genomes were reconstructed and growth rates estimated, so that sequencing data could be used as a relevant, and thoroughly described ground truth to the FISH-FCM data. In fact, from the title of the article, one would expect more insights about the precise kind of phenotypic information that would be exploited in the analysis of microbiomes, whereas the dependence to 16S data is eluded. In a revision, the fact that the results consist of a comparison with 16S would need to be highlighted.

Since the whole research presented here is based on comparing FISH-FCM to 16S sequencing data, the presented 16S results should also be documented in sufficient detail for a comprehensive assessment of the FISH-FCM results. The authors acknowledge in the Introduction that "(omics) methods [are] prone to technical biases, such as extraction and PCR bias". Thus, the choice of 16S data as sole reference is questionable given biases inherent to this method. In fact, background information about the potential biases and notably at detecting the taxa for which probes have been designed, should be reviewed and provided more extensively.

I see no flaw with methods, experimental data, and results, except that some methodological details are not provided comprehensively enough within the paper, and notably about the 16S experiment. Besides, there is also a lack of info about the controlled experiments from which the results derive, which makes it difficult to understand the systematic 16S vs FISH-FCM comparisons. Indeed, two different experiments (bioreactor and SHIME) are performed after different fixation and labelling assays, which are not described in the workflow proposed in Fig 1. Thus, it is hard to follow between methods and results sections.

Abstract and Introduction

The abstract seems out of sync with the results and makes statements that are either unclear or important but undocumented (incl. methodological). I miss the single-cell results, as all results deal with community-level analysis or single taxa (genus levels). This is also strongly alluded to in the introduction (l.126): "Consequently, phenotypic fingerprinting could be performed per taxonomic group, gathering single-cell information of this specific group within a population." Thus, I suggest irrelevances are trimmed and replaced by info about the presented research, and critically to explain how the methodological assessment fully relies on comparisons to 16S fingerprinting as a reference, using which beta diversity metrics and which

differences between which experiment treatments. The underlying assumptions about 16S as ground truth and about the experimentally-generated populations tested to evaluate the approaches abilities at discriminating treatments, must be introduced in the abstract as well as the expected value of using "two completely different ecosystems were considered" for evaluating the accuracy of the method.

We have adjusted the abstract to accommodate your remarks to the best of our knowledge. We hope these clarifications help.

•

l.57: "The two examples show that the described technique is versatile": I assume these two examples refer to SHIME and bioreactor? If so, could high versatility refer to the ability to for FISH and FCM to run on samples derived from such experiments? if so, than this would not be a new demonstration as this methods have been applied to multiple ecosystems. Or, is versatility referring to the ability to detect differences between different fixation and staining treatments (as for the 16S ground truth)? If so, the results are in fact more nuanced as some treatments do not reveal differences. Hence, if versatility here is about using any sort of fixation or staining, then please be more accurate in reporting the finding of which protocol works best at recovering the patterns observed using 16S.

The versality we talk about is the fact that it can be used for widely different microbial community types (simulated gut versus trickling filter on plastic).

- **The sentence l.128 is not clear and tautological. "monitor the response of bacteria down to small process differences" is not clear: what sort of process? and what differences?; remove "thus adding more detail" as it is potentially and not clear (more detail can be more noise so more info about the kind of details and how useful these would be is necessary); "focussing on the taxonomic groups playing a major functional role" and the "monitor[ing] the response of bacteria down to small process differences" is .**

This sentence is an hypothesis but what it states is very general and already verified as FISH-FCM has already known to perform well. A more precise hypothesis related to the work performed and that consists in a systematic comparison with 16S would need to the reformulated.

Thank you for your feedback. we have revised the sentence to clarify the type of process differences being monitored and removed the ambiguous phrase "thus adding more detail." The revised sentence now specifies that the technique focuses on taxonomic groups with significant functional roles and highlights the systematic comparison with 16S rRNA gene sequencing to validate its effectiveness. we hope this addresses your concerns:

"The combined technique could enable us to monitor bacterial responses to specific process variations, focusing on taxonomic groups with significant functional roles. This

approach allows for a systematic comparison with 16S rRNA gene sequencing to validate its effectiveness.”

- **L134: one objective was that "FISH probe efficiency was investigated", but I struggle to see a thorough report on the efficiencies of each of the 10 probes presented on Suppl table 1.**

Thank you for pointing this out, as we used well established probes from literature, we only performed high level efficiency tests (i.e. tested if the probes bound significantly to the cells). We have adjusted the objectives a bit to better comprise what was performed during the study as follows:

“FISH probe efficiency was **confirmed**, the microbial **beta**-diversity and **its** dynamics in the different ecosystems were calculated and accuracy for sample-level classification was assessed compared to 16S rRNA amplicon sequencing.”

- **L134: what is actually performed should be details instead of the more general "the microbial diversity and dynamics of the different ecosystems were calculated". Indeed, no alpha diversity indices was calculated or no analysis calculating statistics based on the time component was performed. In fact, only beta diversity differences using bray-curtis, relative abundance comparisons and permutation tests were done. This should be clear. Note that using relative abundances for compositional data such as 16S amplicon sequencing has shortcomings that would need to be solved or discussed since the 16S serves as ground truth: see Gloor et al. (2017 <https://doi.org/10.3389/fmicb.2017.02224>) and Quinn et al. (2019 <https://doi.org/10.1093/gigascience/giz107>)**

Thank you for pointing this out, as discussed above, we have adjusted the objectives to better and more detailed state what was actually performed as follows:

“FISH probe efficiency was **confirmed**, the microbial **beta**-diversity and **its** dynamics in the different ecosystems were calculated and accuracy for sample-level classification was assessed compared to 16S rRNA amplicon sequencing.”

We agree with you that compositional data comes with its shortcomings, but when read numbers are similar and thus sequencing depth is similar (which is the case here) over the different samples, it is widely accepted that it can be used for relative abundances.

Methods

- **The methods need revision for as non-experts in both FISH and flow cytometry that I believe could both be interested. Useful info for comprehension is missing, some methodological details need clarification, some concepts that are obvious to flow cytometry experts need to be demystified for non-experts, and to understand the**

whole procedure and goals, analytical choices must be justified with more accurate information from cited literature, which is particularly true for the chosen probes and the FCM analysis parameters.

Response: Thank you for this comment, to keep it short and concise we tried to stick to the information needed to recreate the experiments, however we can see the benefit of explaining the choices made in some instances.

We adjusted to text in the material and methods for the FISH part as follows:

“Established FISH probes were chosen based on their ability to bind with taxonomic groups present in the samples, binding effectiveness in literature and/or presence in Probasebase ...”

Within the supplementary information you can find a list of all the probes used and from what reference they were taken (SI table 1). The cross reference to this table is also found in the main text.

The flow cytometry was run at standard conditions, as discussed in lines 150-160. We added in some information on cell concentrations and have put the voltage settings of the detectors in the SI. The fluorophores used were chosen based on the detectors and flow cytometer machines available. Gating strategy can be found in the SI as well (SI figure 6).

- **The "Sampling" section could summarise the sampling units and terms associated with them (which samples are fixed how, and their sampling times) to more easily understand which methods connect to which results presented in the text and figures, as a way to present the complete workflow, as it is not true that (l.169) "The complete proposed workflow is shown in Fig. 1." Please consider expanding Fig.1, or providing a table presenting the samples and their features in a systematic way e.g., with columns such as "origin" ("bioreactor effluent" or "synthetic gut"). In particular, this figure/table details should introduce the names of the "groups" classified by the random forest tests as what is attempted is unclear.**

We have adjusted the Sampling section to have a more clear concept of the types of samples taken. We hope this clarification helps. More information on this can be found in the SI.

“Sample collection and experimental setups of both ecosystems are described in supporting information SI. In short, marine ecosystem samples were collected from the effluent of four trickling filter bioreactors: two with PHBH plastic (referred to as P) and two with a novel plastic, B4PF01 (referred to as F), to investigate plastic biodegradation. Effluent samples for fixation and flow cytometry were collected thrice weekly for 56 days. Samples for 16S rRNA gene amplicon sequencing were collected on days 0, 14, 28, 35, 42, and 56. In vitro simulated gut microbiome samples were derived from the Simulator of the Human Intestinal Microbial Ecosystem (SHIME, Prodigest, Zwijnaarde, Belgium). The SHIME model simulated the proximal and distal colon in different, but consecutive, vessels, with controlled pH, residence time, temperature (37°C), mixing (200 rpm), and diet (Minnebo et

al., 2021; Molly et al., 1993; Van de Wiele et al., 2015). Two setups were used, each in duplicate. The first was a standardized SHIME with fixed eating patterns, transit time and nutritional media. The second was an individualized SHIME with adjusted parameters, according to faecal donor characteristics. Faecal donor samples were inoculated in the colon vessels, and the system stabilized over 11 days. Samples were collected on days 1, 2, 4, 7, 9, and 11 for analysis (FCM and 16S rRNA gene amplicon sequencing) (Minnebo, 2023)."

- **In this last machine learning section are referred the "different staining methods in both ecosystems" and so it would be good to clarify what are all these in the "Sampling" part as a reader must be able to grasp all experimental design aspects in the main manuscript. Hence, please move the supplement info about sampling into the main document.**

We added in a table to clarify what type of samples and what staining methods were used for the samples. We hope this helps for the clarification.

- **I question the relevance of "marine" samples. First, marine ecosystems are highly heterogeneous and so it is not clear whether these are water, sediment or any other substrate, and thus, how were the probes selected for. Also, it seems that "marine ecosystem samples originated from the effluent of four trickling filter bioreactors". What would "originated" means? Are the samples bioreactor effluents discharged into and then sampled from the marine environment? Or, as I suspect, were these marine samples added into the trickling filters, and then the effluent of the reactors were sampled? Please revise according, e.g. "Four marine <sample type> samples were added into trickling filter bioreactors: two containing the plastic poly(3-hydroxybutyrate-co-3-hydroxyhexanoate) (PHBH) and two containing a novel plastic of PI-protected chemical composition (hereafter B4PF01)". Could "P" and "F" reactors be defined here instead of being re-defined later again in the Results sections?**

Thank you for your comments. We acknowledge the need for clarity regarding the "marine" samples. The samples in question were, as you were suggesting, seawater samples (i.e. coastal water taken at the Belgian coast in Ostend) enriched on plastic in a trickling filter. Effluent samples were further analysed in this study. The terminology 'marine' samples was used to have a clear and easy distinction between the two environments (gut microbiome and marine microbiome). We acknowledge the heterogeneity of the marine, and in general, natural environments, which is why we choose the higher-level taxonomic groups as targets for FISH.

We have also addressed P and F reactors in the methods section:

"Marine ecosystem samples were collected from the effluent of four trickling filter bioreactors: two with PHBH plastic (referred to as P) and two with a novel plastic, B4PF01 (referred to as F), to investigate plastic biodegradation."

- Because of the high amount of methods and reagents used, the reading is complicated by frequent sentence interruptions by in-parentheses versions, brand/catalogue info, wavelengths, acronyms and, my above concern, references to sample types and experimental treatment that are only described the missing "Sampling" methods section. While I applaud the level of details about the material used and steps involved in FISH and FCM, I suggest the author consider evacuating all these highly technical info into a STAR methods table (see <https://www.elsevier.com/researcher/author/tools-and-resources/key-resources-table>), with sections for the different aspects of the experiments (sampling), fixation, dyes, etc.

We truly appreciate your feedback and understand the complexity introduced by the detailed methods and reagents. While we acknowledge the potential benefits of a STAR methods table, our attempts to implement this approach did not yield significant improvements in readability or clarity. We believe that referencing the detailed information within the text remains essential for the comprehension of our experimental procedures.

- The 16S experiment is based on DNA extracts and not RNA (cDNA). Hence, it should be referred to as "16S rRNA gene" or "16S rDNA" to avoid confusion.

Thank you for your observation. You are absolutely correct. We have revised all mentions to ensure consistent use of the notation '16S rRNA gene amplicon sequencing' throughout the manuscript.

- It is insufficient to state an "in-house phenol-chloroform DNA extraction protocol". More details or a reference describing the extraction procedure in details is needed to allow for review and potential replication.

We have adapted the manuscript to provide more clarity:

“DNA from the marine ecosystem samples was extracted by means of bead beating with a PowerLyzer instrument (MoBio) and phenol/chloroform extraction as described by Van Landuyt et al, (2020). The extraction was followed by PCR with universal bacterial primers (341F (5'-CCT ACG GGN GGC WGC AG-3') and 785Rmod (5'-GAC TAC HVG GGT ATC TAA KCC-3') targeting V3-V4 region of the 16S rRNA gene and send to LGC genomics for Illumina (Van Landuyt et al., 2020). DNA from the SHIME ecosystem samples were extracted and sequenced according to Vandeputte et al, (2017) (Vandeputte et al., 2017)”

- What justifies using tSNE and NMDS in addition to PCoA?

We have chosen to use t-SNE, NMDS, and PCoA in our analysis for the strengths of each technique and also to provide a comprehensive understanding of our data:

t-SNE, specifically, preserves local structures and relationships within the data, making it excellent for visualizing clusters and identifying patterns in high-dimensional data by maintaining neighborhood integrity.

NMDS preserves the rank order of distances between data points, which is useful for visualizing data where the exact distances are less important than the relative ordering, making it robust to non-linear relationships.

And a PCoA preserves the actual distances between data points, providing a global view of the data structure and making it suitable for understanding overall patterns and relationships.

By combining these techniques, we can capture both local and global structure and it allows us to gain a more nuanced and detailed understanding of our data.

Results

- **It is confusing that novel acronyms are introduced here. Please define these in the "Sampling" section of the methods so that there is no need to redefine treatment groups. These should be clear as early as possible as the most important here is for the reader to understand what are the different treatments that are compared and how these comparisons are diagnostic to measuring the improvement offered by the proposed methodological innovation. In fact, this first result paragraph is to be moved entirely to the "Sampling" methods (and potentially summarized as a table with the treatments, sample sizes etc?), before the more details about underlying fixation, hybridization and staining methodologies (which are less important and could also be summarized into a STAR table).**

As per your previous suggestion, we have added the information of the sampling in the material and methods and have simplified it in the results. To explain better why the different staining methods were used we have added the following:

“Using four different staining methods we investigated if the flow cytometric fingerprinting could be improved: (1) non-fixed samples stained with Sybr Green (SG), (2) paraformaldehyde (PFA) fixed samples stained with SG, (3) fixed samples stained with 4',6-diamidino-2-phenylindole (DAPI) and (4) fixed FISH samples counterstained with DAPI.”

- **The results highlight a lack of details in the methods and further results are necessary to understand the reason for the existence of clusters in the 16S results used as reference for the FISH-FCM data, e.g., is there lower alpha diversity in some treatments? I could find no sample and sample-prep metadata information that would indicate the potential effects of extraction or sequencing batch. Also, it is said that negative and positive controls are used but there is no report about these.**

As per your previous comments, we have added in quite some information on the sampling in the material and methods. The extraction was performed the same and at the same time for all samples of the same experiment (so only one batch) (i.e. marine or SHIME). Inter-experiment differences would not matter as we compare within experiment but not across.

Positive (heat-killed samples e.g. for the damaged population) and negative (filtered samples or empty buffer samples) controls were used to set the cell gating strategy for the flow

cytometer. The strategy is explained better in the SI and thus the controls are shown there. They are excluded from the diversity analysis as they do not represent actual samples.

We checked the alpha diversity within each experimental dataset but it did not seem to add value to the final work, so chose to leave it out. If you require us to add, we can.

- **It appears that no actual phenotypic information is being extracted from the multiple FCM experiments, challenging what is announced in the abstract. Is it true that data derived from flow cytometry relies on these features but research aiming to establish improvement of a methods relying on these (as per the title) would be expected to provide in-depth descriptions of the measures phenotypic variation, which is possible as shown is other papers demonstrating the method.**

Although not specifically mentioned, we take into account the FSC and SSC channel as well as sometimes both the area as well as the height of the fluorescent signals, which is in itself phenotypic information. FSC and SSC can be related to cell size and granularity respectively, the signal area vs height or width can give information about the stickiness of the cells (i.e. are cells sticking together) or are the cells more elongated rather than round. Moreover, a nucleic acid dye in itself helps to identify events as cells, but is also able to identify HNA and LNA cell populations, which is also phenotypic information. Inherently, information obtained from a flow cytometer that cannot be directly linked to taxonomy (i.e. FISH probes) is always phenotypic information.

- **In fact, the quality of the FISH-FCM method is measured based on matching to the 16S data, by checking whether the same diversity patterns are obtained, so the assumption is that the 16S result is a good standard, which is not inline with how this is presented in the intro where 16S is explained as being more biased. Hence, one would expect that an improved FISH-FCM method would yield different results, incl. non-significant Mantel correlations. This calls for alternative ground truth data, which is not available here. One solution for ground truthing would be a compelling experiment with known microbial composition and known dynamics (like in Brüwer et al. 2023 mSystems). Since the data stems from experimental inoculates originating from complex environments, such ground truth could be achieved but then the experiments must be described in great detail in the main document and additional about the 16S read abundances and diversity should be presented. For the bioreactors, however, it appears that a plastic degradation process is being referred to as a driver of community dynamics, but this process is not introduced or described and thus fails to convinced as having ground truth value.**

Thank you for your comment, we have added in additional information on the process in the sampling part of the material and methods, which should offer you insight in why we assume the plastic degradation is the driver of these communities.

- The process of monitoring plastic biodegradation in the reactors is mentioned throughout the manuscript, incl. Abstract, Intro, and also in the Results, when it comes as a new goal (l.242 "to clarify the plastic degradation process"). However, this issue is not documented by any result, and in the Results section, it appears as a plain discussion element with references (l.334), which should be moved to the Discussion section, or removed altogether as the plastic aspect seems irrelevant. I recommend either expanding on the significance of the plastic degradation process with insightful data and results in relation to your fixation treatments and FISH-FCM findings, or removing it (incl. in suppl methods "set up to investigate plastic biodegradation (unpublished data)"). Alternatively, this could be moved to the methods to justify the probes choice instead of "FISH probes were chosen based on literature and Proebas"?

We have adjusted the text a bit, the reactors were set-up to identify and clarify plastic degradation of different types of plastic. However, within the context of this paper, the experimental set-up of the reactors facilitate the possibility to see if two communities run in the same set-up given a similar carbon source (plastic) but differently made (PHBH vs. other bioplastic) would drive to a different microbial community.

- Please stick to describing results in the "Results" section and avoid adding experimental reasoning and justifications at this point. For example, the fact that different plastics may be/are different carbon sources in the experiment should be introduced before if relevant in the results, either in the introduction. Whether communities separate because of the presence of plastics, and whether these plastics represent different carbon sources may be a discussion element, but certainly not fall in the midst of the results. Moreover, for this aspect to be discussed properly, it would be necessary to know the plastic dosage and the presence of other, less refractory carbon sources that are likely present in bioreactors. Then, more info about the reactors would be necessary (e.g., pre-treatment) as well as about the sample (sediment? water?).

We have added in the relevant information in the material and methods as per your previous comment and feel like the information now given in the results to distinguish between the two different trickling filters does not add new information to the text, rather is just used as a tool to follow which result is linked to which sample.

- Also, I would like to argue that "FISH staining" is inappropriate as FISH is not a staining method per se because it is based on the hybridization of fluorescent probes, in contrary to Sybr Green, propidium iodine and DAPI.

Thank you for your feedback. We have renamed FISH staining to FISH labelling throughout our manuscript.

- Importantly, I suggest that the cited research is cited in a more accurate and relevant manner in the Introduction statements. Here are two examples:

- 1.80: "On the other hand, genotypic information has, up to now, been obtained mostly by more time-consuming and expensive (omics) methods, prone to technical biases, such as extraction and PCR bias, and without absolute microbial abundance quantification (Özel Duygan & van der Meer, 2022; Props et al., 2016a)." Özel Duygan & van der Meer (2022) is about the potential and applications of machine learning to extract clusters and knowledge from FCM data alone, and only invokes omics as follow up tech, or in their introduction as powerful tools yet requiring complementary data such as FCM. This referenced work does not present actual results on costs, technical biases, or the ability of omics to infer microbial abundances. However, it is cited 7 times in the intro, incl. at places where it seems irrelevant (it is not dealing with "classical diversity analysis based on diversity indices like the Hill numbers")

Thank you for your remark, we have adjusted our citing to more accurately and relevantly convey our points.

- 1.83: "However, more recently, big strides have been made to overcome PCR bias and time constraints, with for example the Oxford Nanopore Technology (ONT) (Zorz et al., 2023), where, in theory, (16S rRNA) gene sequencing of the community can be performed within a day's work without amplification". Here, Zorz et al. (2023) is about a portably ONT sequencing solution also relies on (full SSU) PCR, and thus would also be biased as for the standard Illumina 16S rRNA PCR protocol - which is what the comparison between these two tech actually show. In fact, I questions the very relevance of this sentence since the authors' work do not rely on long-range PCR or long-read sequencing. Overall, I encourage the authors to read the cited works more carefully to only extract relevant results, that actually support statements, as it may be misleading for the readership.

Thank you for your remark, we have adjusted our citing to more accurately and relevantly convey our points and we acknowledge that as soon as you talk about 16S amplicon sequencing you need to do PCR (to make the amplicon).

Minor comments:

- **Citation are not formatted properly (redundant), e.g.**
 - "[...] described by Van Landuyt et al, (2020) (Van Landuyt et al., 2020)." [...] according to Vandeputte et al, (2017) (Vandeputte et al., 2017)." Or confusing: - 1.207: "Rubbens et al. (2020) (Rubbens et al., 2021c)"
 - These citations are duplicated: -- Rubbens, P., Props, R., Kerckhof, F.-M., Boon, N., & Waegeman, W. (2021a). Cytometric fingerprints of gut microbiota predict Crohn's disease state. *The ISME Journal*, 15(1), 354-358. <https://doi.org/10.1038/s41396-020-00762-4>
 - Favere, J., Buyschaert, B., Boon, N., & De Gussemé, B. (2020a). Online microbial fingerprinting for quality management of drinking water: Full-scale event detection. *Water*

The citation to Van Landuyt et al. 2023 justifying the taxonomic specificity of FISH probes is not referenced properly and is not accessible. If the thesis can be made accessible, then it would also be good to provide context about how the experimental conditions in this thesis have demonstrated genus-level FISH probing relevant to the proposed methodological improvement, e.g., was there consistence across many samples? same PFA fixation? It is impossible to access and thus ascertain relevance.

We have checked the citing and adjusted it to be more careful, thank you for your help with detecting these mistakes.

- **l.144: "diversity analysis" (no hyphen)**

Thank you for finding this mistake. We have adjusted it in the manuscript.

- **l.196: "relatively large number of cells in a community": relative to what?**

We understand the confusion and have decided to remove this section to enhance readability.

- **l.199: "(which can be done by different mathematical approaches)": either remove or describe some of the most useful approaches, to demystify such advances to an interested readership.**

Thank you. We have decided to exclude it from the manuscript.

- **l.103: "(for example for determining the Crohn's disease state and monitoring drinking water quality (Favere et al., 2020a))": this citation of an accurate result could be a whole sentence without parentheses as such successful applications are relevant. However, this reference is about drinking water, not about determining the Crohn's disease state. Please add the reference about determining the Crohn's disease state: if this is "Rubbens et al., 2021a", please cite at the right place.**

Thank you for your feedback. We have updated our manuscript to enhance readability:

“This fingerprint has been shown to be indicative of taxonomic changes in microbial community composition dynamics. **For example, it has been used to determine the Crohn’s disease state and to monitor drinking water quality** (Favere et al., 2020; Rubbens et al., 2021a). **Additionally**, it can be deployed for classical diversity analysis based on diversity indices like the Hill numbers (Chao et al., 2010; Özel Duygan & van der Meer, 2022; Props et al., 2016).”

- **l.247: "Principal Coordinates Analysis (PCoA) was performed to diversify between microbial community fingerprints": it is not clear what " diversify" means in this context. Would you the author mean "distinguish"?**

You are correct. We have adjusted the manuscript accordingly:

“Principal Coordinates Analysis (PCoA) was performed to **distinguish** between microbial community fingerprints (based on Gaussian Mixture Models (GMM)) growing on the two types of plastics (F vs. P).”

Figures:

- **A comparison on Figure 4 is made across reactors, and so only then are introduced the reactors names and sample sizes. Please add this info on an expanded Fig1 clarifying the full experiment.**

The information you requested (as discussed in previous remarks) was added in the sampling part of the material and methods, however we don't see why we would add this to figure 1 as we would prefer this to be a general figure that could work for any experimental set-up not only the two examples we tested/validated the method on.

I suggest that time points on the ordinations are colored using a gradient color scheme as there seem to exist interesting change dynamics in the data.

We decided on these colors to make the differences clear, so we have decided to keep the colors as is.

Supplement:

- **l.17: replace "The proximal and distal colon pH of 5.6-5.9, respectively, 6.6-6.9, was maintained with built-in pH controllers and pumps regulating the dosage of 0.5 M NaOH and HCl (Chem Lab, Zedelgem, Belgium)." by "The pH of proximal and distal colon vessels was kept within 5.6-5.9 and 6.6-6.9, respectively using controllers and buffer pumps (Chem Lab, Zedelgem, Belgium)." If pH ranges are given, is it relevant to indicate that there are "pH controllers and pumps regulating the dosage of 0.5 M NaOH and HCl"?**

We respectfully don't think it is a problem to specify that the controllers are there for the pH.

- **The figure captions could remind which experiments one being looked at (SHIME vs bioreactor) and which treatment therein, even though this is evident from the legend, i.e., please elaborate figure too so the presented results refer to the rather complicated experimental design more clearly.**

We have added in either “**marine trickling filter samples**” or “**simulated gut samples**” to all the relevant figure captions.

- **Similarly, please add titles to the Suppl tables 3 and 4 for clarity: both start with Supplementary Table 3 - By means of three different statistical tests, the multivariate data was analysed. The Bray-Curtis dissimilarity matrix was used to test the distance between the centroids of the groups with different plastics (F and P)**

We have adjusted the captions of the supplementary tables you mentioned by adding in either **“marine trickling filter samples”** or **“simulated gut samples”**.

Re: Spectrum01973-24R1 (Integrating Taxonomic and Phenotypic Information through FISH-enhanced Flow Cytometry for Microbial Community Dynamics Analysis)

Dear Prof. Nico Boon:

Your manuscript has been accepted, and I am forwarding it to the ASM production staff for publication. Your paper will first be checked to make sure all elements meet the technical requirements. ASM staff will contact you if anything needs to be revised before copyediting and production can begin. Otherwise, you will be notified when your proofs are ready to be viewed.

Sincerely,
Adriana Lopes dos Santos
Editor
Microbiology Spectrum